# Exploiting horizontal pleiotropy to search for causal pathways within a Mendelian randomization framework

Yoonsu Cho [1], Philip C. Haycock[1], Eleanor Sanderson [1], Tom R. Gaunt [1], Jie Zheng[1], Andrew P. Morris[2,3], George Davey Smith [1] & Gibran Hemani [1✉]

In Mendelian randomization (MR) analysis, variants that exert horizontal pleiotropy are typically treated as a nuisance. However, they could be valuable in identifying alternative pathways to the traits under investigation. Here, we develop MR-TRYX, a framework that exploits horizontal pleiotropy to discover putative risk factors for disease. We begin by detecting outliers in a single exposure–outcome MR analysis, hypothesising they are due to horizontal pleiotropy. We search across hundreds of complete GWAS summary datasets to systematically identify other (candidate) traits that associate with the outliers. We develop a multi-trait pleiotropy model of the heterogeneity in the exposure–outcome analysis due to pathways through candidate traits. Through detailed investigation of several causal relationships, many pleiotropic pathways are uncovered with already established causal effects, validating the approach, but also alternative putative causal pathways. Adjustment for pleiotropic pathways reduces the heterogeneity across the analyses.

[1] MRC Integrative Epidemiology Unit, Population Health Sciences, Bristol Medical School, University of Bristol, Bristol BS8 2BN, UK. [2] Department of Biostatistics, University of Liverpool, Liverpool L69 3GL, UK. [3] Division of Musculoskeletal and Dermatological Sciences, University of Manchester, Manchester M13 9NT, UK. ✉email: g.hemani@bristol.ac.uk

Mendelian randomisation (MR) is now widely used to infer the causal influence of one trait (the exposure) on another (the outcome)[1,2]. It generally uses genetic instruments for an exposure, obtained from genome-wide association studies (GWAS). If the instruments are valid, in that they are unconfounded and influence the outcome only through the exposure (vertical pleiotropy), then they will each provide an independent, unbiased estimate of the causal effect of the exposure on the outcome[3]. Meta-analysing these estimates can provide a more precise estimate of the effect of the exposure on the outcome[4,5]. If, however, some of the instruments are invalid, particularly because they additionally influence the outcome through pathways that bypass the exposure (horizontal pleiotropy)[3], then the effect estimate is liable to be biased. To date, MR method development has viewed horizontal pleiotropy as a nuisance that needs to be factored out of the analysis[6–9]. Departing from this viewpoint, here we exploit horizontal pleiotropy as an opportunity to identify alternative traits that putatively influence the outcome. We also explore how this knowledge can improve the original exposure–outcome estimates.

A crucial feature of MR is that it can be performed using only GWAS summary data, where the effect estimate can be obtained solely from the association results of the instrumental single-nucleotide polymorphisms (SNPs) on the exposure and on the outcome[5]. This means that causal inference between two traits can be made even if they have never been measured together in the same sample of individuals. Complete GWAS summary results have now been collected from thousands of complex trait and common diseases[10], meaning that one can search the database for candidate traits that might be influenced by SNPs exhibiting possible pleiotropic effects (outliers). In turn, the causal influence of each of those candidate traits on the outcome can be estimated using MR by identifying their instruments (which are independent of the original outlier). Should any of these candidate traits putatively influence the outcome then this goes some way towards explaining the horizontal pleiotropic effect that was exhibited by the outlier SNP in the initial exposure–outcome analyses.

Several methods exist for identifying outliers in MR, each likely to be sensitive to different patterns of horizontal pleiotropy. Cook's distance can be used to measure the influence of a particular SNP on the combined estimate from all SNPs[11], identifying SNPs with large influences as outliers. Steiger filtering removes those SNPs that do not explain substantially more of the variance in the exposure trait than in the outcome, attempting to guard against using SNPs as instruments that are likely to be associated with the outcome through a pathway other than the exposure[12]. Finally, meta-analysis tools can be used to evaluate if a particular SNP contributes disproportionately to the heterogeneity between the estimates obtained from the set of instruments, and this has been adapted recently to detect outliers in MR analysis[13–15]. A potential limitation of heterogeneity-based outlier removal is that this practice could constitute a form of cherry picking[9,16]. While outlier removal can certainly improve power by reducing noise in estimation, it could also potentially induce higher type 1 error rates, which we go on to explore through simulations.

Recent large-scale MR scans have indicated that horizontal pleiotropy is widespread based on systematic analysis of heterogeneity[14,17]. This suggests that many SNPs used as instruments are likely to associate with other traits, which in turn might associate with the original outcome of interest—hence giving rise to heterogeneity. As such we have an opportunity to identify previously unreported pathways by making use of outlying instruments in an MR analysis. Equipped with automated MR analysis software[10], outlier detection methods and a database

of complete GWAS summary datasets, we develop MR-TRYX (from the phrase 'TReasure Your eXceptions'[18], popularised by William Bateson, an early pioneer in genetics). This framework is designed to identify putative causal factors when performing a simple exposure–outcome analysis.

In this paper, we describe how MR-TRYX can be implemented in MR analyses and how to interpret its results. A wide range of simulations is performed to show how knowledge of horizontal pleiotropic pathways can be used to improve the power and reliability of the original exposure–outcome association analysis. Our simulations also show that outlier removal methods can induce bias or increase type 1 error rates, but adjustment for detected pleiotropic pathway can improve estimates by reducing heterogeneity without sacrificing study power. We apply MR-TRYX to four exemplar analyses to demonstrate its potential utility, showing that horizontal pleiotropic pathways can be used to discover putative causal factors for an outcome of interest.

## Results

**Overview of MR-TRYX.** Figure 1 shows an overview of the approach. MR-TRYX is applied to an exposure–outcome analysis in a two-sample MR framework and it has two objectives. The first is to use outliers in the original exposure–outcome analysis to identify putative factors that influence the outcome independently of the exposure. The second is to re-estimate the original exposure–outcome association by adjusting outlier SNPs for the horizontal pleiotropic pathways that might arise through the putative associations. This outlier-adjustment method should be treated as a new approach to be used in conjunction with other methods that already exist in the MR sensitivity analysis toolkit. We provide extensive discussion on the context, advantages and potential pitfalls that come with trying to use a data-driven approach to adjust for horizontal pleiotropy at the end of the paper.

**Adjustment of pleiotropic pathways improves MR performance.** We performed a wide range of simulations (Fig. 2, Supplementary Data 2) to evaluate how a variety of methods designed to deal with pleiotropy fare under a set of different scenarios that violate the exclusion restriction principle. Perhaps the most striking result from these simulations is that no method is always reliable, and several methods have similar overall reliability while performing very differently from each other between specific scenarios. Across 47 simulation scenarios, adjusting for detected outliers using the MR-TRYX framework had the highest average rank, and simply performing inverse-variance weighted (IVW) random effects was most often the best performing method, whereas removing detected outliers had the lowest average rank. We note that generally we do not know which of the scenarios are of relevance for any particular empirical analysis and so the metric used to evaluate performance here reflects the methods that are most generally performant. We found that as the proportion of instruments exhibiting pleiotropic effects increased, all methods typically worsened in their performance though there were notable examples in which increasingly widespread pleiotropy does not have an adverse effect. For example, widespread balanced horizontal pleiotropy or mediated pleiotropy does not have a drastic adverse influence on IVW, and MVMR and outlier adjustment is relatively impervious to confounding pleiotropy.

It is an obvious conceptual disadvantage in these simulations for IVW and outlier removal, which use only the exposure and outcome data, when compared against MVMR and MR-TRYX which draw on information from other sources. However, we note that the MR-TRYX adjustment approach depends on detecting candidate traits that explain the pleiotropic effect and if the relevant candidate traits are not available, there is no

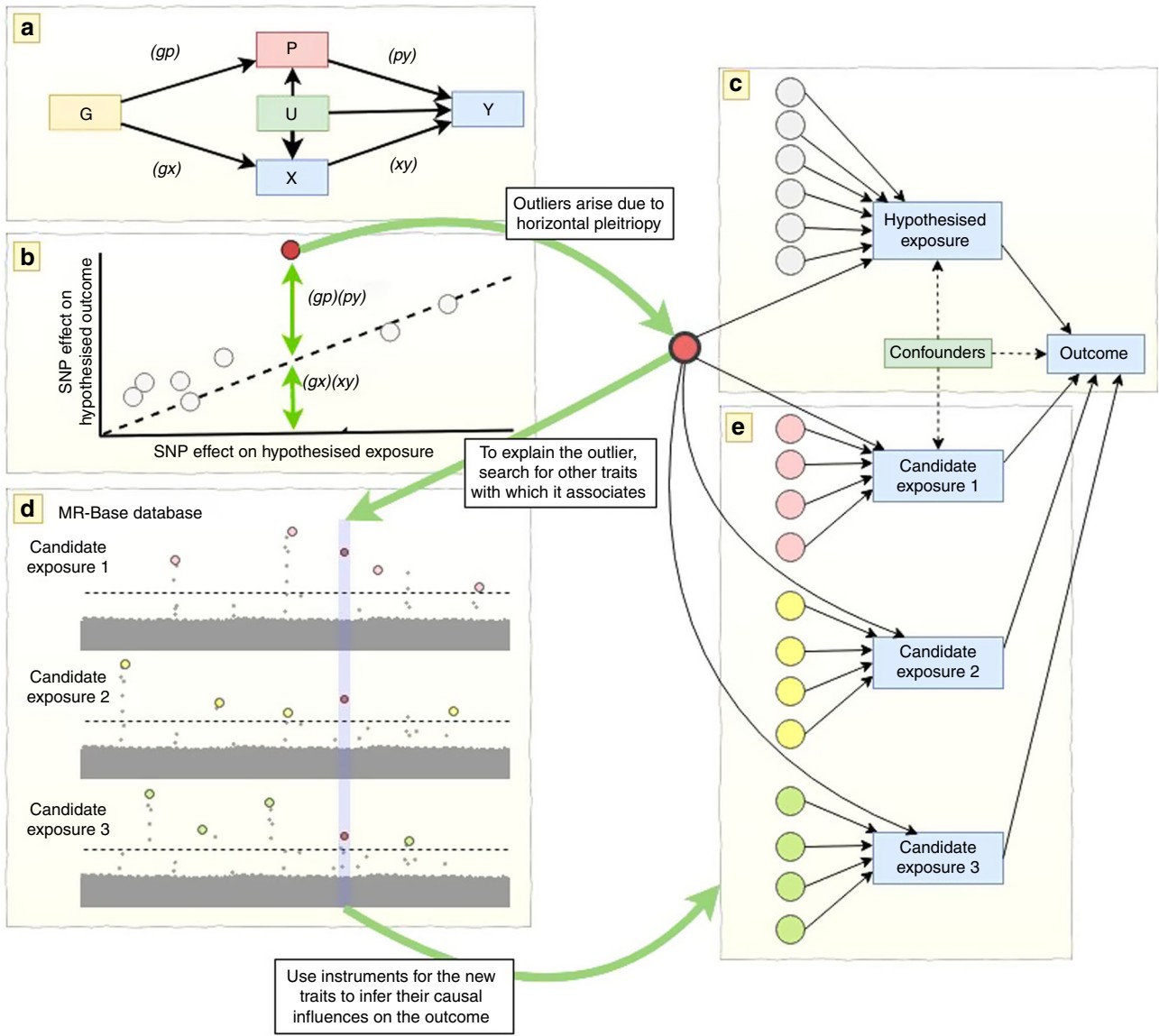

**Fig. 1 Conceptual framework of the study.** Illustration of identifying putative factors that influence the original observations. **a** Where (*gx*) is the SNP–exposure effect, (*xy*) is the exposure–outcome effect as estimated through MR analysis from the non-outlier SNPs, (*gp*) is the SNP–candidate trait effect and (*py*) is the causal effect of the candidate trait on the outcome. **b** The open circles represent valid instruments and the slope of the dotted line represents the causal effect estimate of the exposure on the outcome. The closed red circle represents an outlier SNP which influences the outcome through two independent pathways, P and X. **c** One way in which the red SNP can exhibit a larger influence on the outcome than expected given its effect on the exposure is if it influences the outcome additionally through another pathway (horizontal pleiotropy). **d** Using the MR-Base database of GWAS summary data for hundreds of traits we can search for 'candidate traits' with which the outlier SNP has an association. **e** Instruments excluding the original outlier SNP are obtained for each candidate trait, LASSO-based multivariable MR is used to prune the candidate traits to avoid redundancy, and the causal influence of each of those independent candidate traits on the outcome can subsequently be estimated. This allows us to identify alternative traits that putatively influence the outcome and adjust the SNP-outcome associations for pleiotropic pathways in the original exposure-outcome model.

adjustment and the method becomes identical to random effects IVW which generally performs better than outlier removal. We also note that if we use association with candidate traits to determine whether or not to remove an outlier, then improvements can be made over simple outlier removal. What we observe here is intuitive because the potential drawback of outlier removal is that the outliers could be the only valid instruments, or false discovery rates increase due to overly precise confidence intervals. Thus, adding an extra barrier to the removal of outliers can mitigate these problems.

Multivariable MR targets a different estimand than univariable MR—it is estimating the direct effect rather than the total effect of

the exposure on the outcome. This strategy performs generally well across the range of simulations except in the case when the candidate trait is a mediator of the x–y association in which case there is a strongly attenuated direct effect. The problem here is that it is hard for MVMR to distinguish between a model where the exposure's influence on the outcome is mediated by a candidate trait (the exposure is causal), vs. where the exposure's apparent effect on the outcome is simply due to pleiotropy through the candidate trait (the exposure is not causal)[19]. Here, MVMR performs worse than other methods when the candidate trait is a mediator, as MVMR estimates the direct effect of x on y adjusting for the entirety of x's signal on y. Adjusting for outliers

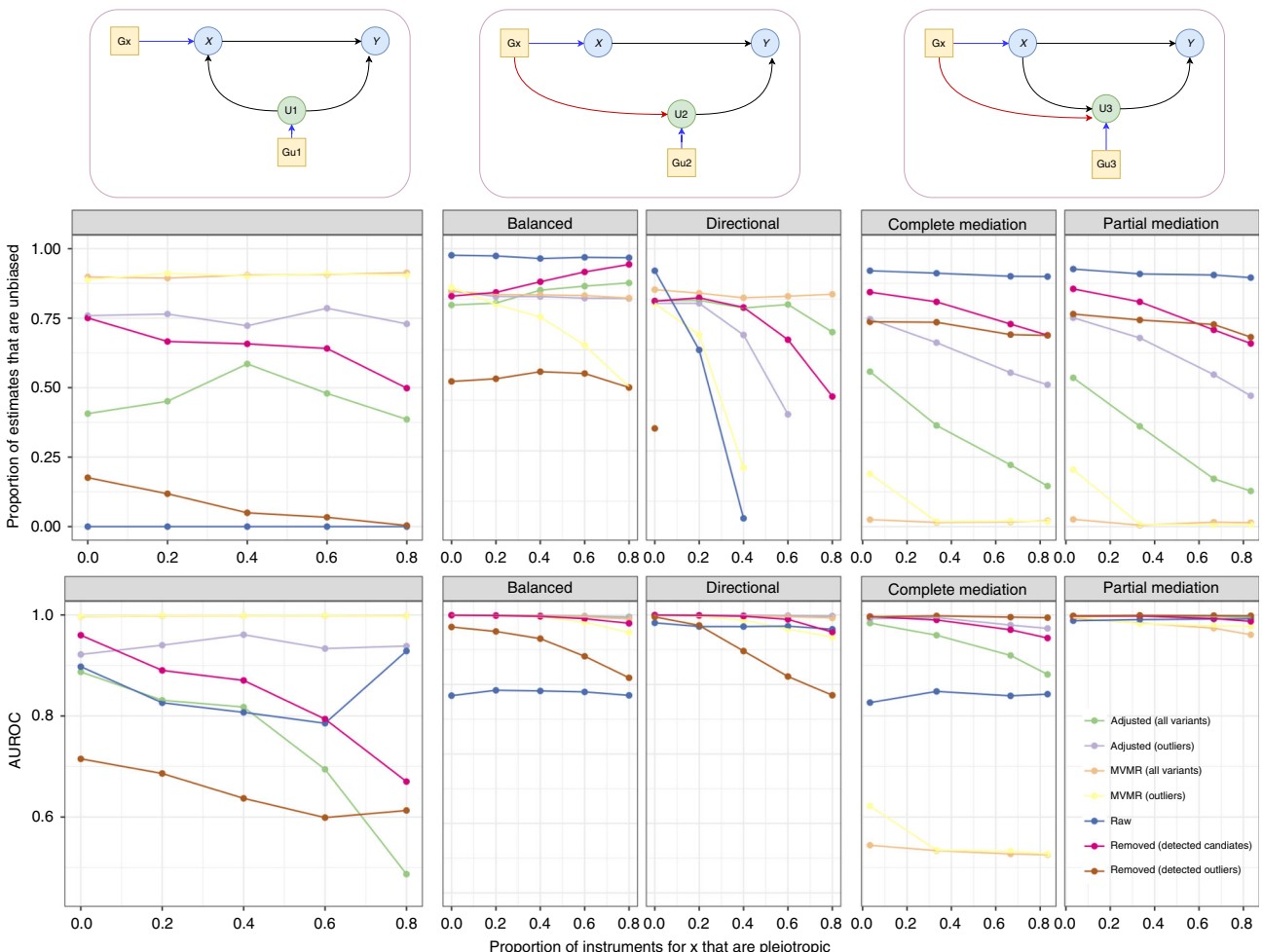

**Fig. 2 Simulations comparing methods across different scenarios.** We evaluated three scenarios: confounding pleiotropy, horizontal pleiotropy and mediated pleiotropy (columns of graphs, with DAGs illustrating the scenarios. See Methods for full details). The x-axis of each graph represents the proportion of variants used to instrument x that were similated to exhibit pleiotropic effects. Typically, 30 instruments were simulated directly for x but this varies across scenarios where necessary. The y-axis of the first row of graphs represents the proportion of simulations that lead to unbiased effect estimates of x on y. The y-axis of the second row of graphs represents the sensitivity and specificity of the analysis across the simulations, where the area under the receiving operating curve (AUROC) represents the ability of the method to distinguish between simulations in which the causal effect of x on y is either null or not null. For all graphs, higher y-axis values are better. Seven methods are evaluated at each simulation. Raw = IVW random effects estimates applied to all detected instruments; Removed = either all outliers are removed, or only outliers detected to associate with a candidate trait; MVMR = multivariable MR using either candidate traits detected to associate with any instrument or using only candidate traits associated with outlier instruments; Adjusted = adjusting SNP–outcome associations for candidate traits applied either only to variants detected to be outliers, or all variants regardless of outlier status.

escapes this problem to some extent because it only adjusts some proportion of the instruments for x that are most likely to be pleiotropic, allowing some signal of x on y to persist due to the unadjusted variants.

**Empirical MR-TRYX analyses.** To examine the performance of MR-TRYX analysis, we tested four independent exposure–outcome hypotheses: (i) systolic blood pressure (SBP) and coronary heart disease (CHD); (ii) urate and CHD; (iii) sleep duration and schizophrenia; and (iv) education level (years of schooling) and body mass index (BMI). For each analysis we: (a) obtain MR estimates of the exposure–outcome causal relationship and detect outlier instruments; (b) identify putative causal influences (candidate traits) on the outcome trait based on their associations with outlier variants (Table 1, Supplementary Data 1); (c) adjust the original SNP–outcome estimates for the putative influences operating through the candidate traits (Table 2); and (d) compare the changes in heterogeneity in the MR estimates of the adjusted SNP–outcome effects to standard outlier removal methods.

Example 1: Systolic blood pressure and coronary heart disease: Blood pressure is a well-established risk factor for CHD. Random effects IVW estimates indicated that higher SBP is causally associated with higher risk of CHD (odds ratio [OR] per 1 SD: 1.76; 95% CI: 1.47, 2.10). While there was substantial heterogeneity in this estimate ($Q = 682.7$ on 157 SNPs, $p = 5.74 \times 10^{-67}$), the estimates from MR-Egger, weighted median and weighted mode methods were consistent (Table 2). Seven of the 157 SNPs were detected as strong outliers based on $Q$ statistics. We identified 69 candidate traits that were associated with these outliers ($p < 5 \times 10^{-8}$). We manually removed redundant traits and traits that are similar to the exposure and the outcome (e.g. hypertension). Among the remaining candidate traits, 15 were putatively causal for CHD (Fig. 3a). After we applied LASSO regression, six traits remained (Table 1): anthropometric measures (e.g. height), lipid levels (e.g. cholesterol level) and self-reported ibuprofen use were among the candidate traits that associated with CHD, which were all uncovered due to two outliers (rs3184504 near *SH2B3* and rs9349279 near *PHACTR*).

**Table 1 Candidate traits associated with both exposure and outcome.**

| Outlier SNPs | Nearest gene | Category | Phenotypes[a] | N SNPs[b] | Beta (95% CI)[c] |
|---|---|---|---|---|---|
| **Empirical analysis 1: systolic blood pressure (mmHg) on coronary heart disease (Odds ratio)** | | | | | |
| rs3184504 | SH2B3 | Early development | Birth weight of first child | 40 | −0.312 (−0.498, −0.126) |
| | | Anthropometric measures | Standing height | 577 | −0.208 (−0.264, −0.152) |
| | | Lipid | LDL cholesterol | 78 | 0.393 (0.290, 0.497) |
| | | | HDL cholesterol | 86 | −0.172 (−0.288, −0.055) |
| | | | Total cholesterol | 86 | 0.378 (0.271, 0.484) |
| rs9349379 | PHACTR | Medications | Self-reported status of ibuprofen intake | 2 | −16.726 (−37.262, −3.811) |
| **Empirical analysis 2: urate (mg/dl) and coronary heart disease (Odds ratio)** | | | | | |
| rs653178 | ATXN2 | Early development | Birth weight of first child | 31 | 0.347 (0.065, 0.628) |
| | | | Birth weight | 40 | −0.312 (−0.498, −0.126) |
| | | Anthropometric measures | Comparative height size at age 10 | 357 | −0.248 (−0.342, −0.154) |
| | | | Hip circumference | 275 | 0.131 (0.030, 0.231) |
| | | | Impedance of arm (left) | 305 | −0.263 (−0.380, −0.145) |
| | | | Standing height | 577 | −0.208 (−0.264, −0.152) |
| | | Lipid | HDL cholesterol | 78 | 0.393 (0.290, 0.497) |
| | | | LDL cholesterol | 86 | −0.172 (−0.288, −0.055) |
| | | | Total cholesterol | 86 | 0.378 (0.271, 0.484) |
| | | Disease | hypothyroidism/myxoedema (Self-reported) | 77 | 0.847 (0.211, 1.483) |
| | | Smoking | Past tobacco smoking | 41 | −0.265 (−0.500, −0.029) |
| | | Medications | Treatment/medication: levothyroxine sodium | 51 | 1.231 (0.270, 2.191) |
| rs642803 | OVOL1 | Anthropometric measures | Waist circumference | 218 | 0.458 (0.352, 0.563) |
| **Empirical analysis 3: sleep duration (hour/night) and schizophrenia (Odds ratio)** | | | | | |
| rs7764984 | HIST1H2BJ | Disease | Malabsorption/coeliac disease (self-reported) | 11 | −8.401 (−12.842, −3.961) |
| rs13107325 | SLC39A8 | Anthropometric measures | Impedance of leg (left) | 282 | 0.179 (0.047, 0.311) |
| | | Memory | Prospective memory result | 2 | 4.493 (1.851, 7.135) |
| **Empirical analysis 4: years of schooling (years) and body mass index (kg/m$^2$)** | | | | | |
| rs6882046 | LINC00461 | Drinking | Alcohol intake frequency | 31 | 0.347 (0.065, 0.628) |
| rs4800490 | NPC1 | Drinking | Alcohol intake frequency | 31 | 0.347 (0.065, 0.628) |
| | | Exercise | Usual walking pace | 22 | −1.595 (−2.364, −0.825) |
| rs8049439 | ATXN2L | Drinking | Alcohol intake frequency | 31 | 0.347 (0.065, 0.628) |

SNP single-nucleotide polymorphism, VLDL very low-density lipoprotein, HDLC high-density lipoprotein cholesterol, LDLC low-density lipoprotein cholesterol, N SNPs number of SNPs, CI confidence interval.
[a]Candidate traits that are associated with outliers ($p < 5 \times 10^{-8}$) and both exposure and outcome are listed. The listed traits were used in the adjusted model to investigate whether they are associated with the hypothesised outcome.
[b]The number of SNPs used for two-sample MR analysis of candidate traits on the outcome.
[c]The results were presented as IVW beta coefficient (95% CI), derived from two-sample MR analyses. Empirical analysis 1: systolic blood pressure (mmHg) and coronary heart disease (log odds); Empirical analysis 2: urate (mg/dl) and coronary heart disease (log odds); Empirical analysis 3: sleep duration (hour/night) and schizophrenia (log odds); Empirical analysis 4: years of schooling (years) and body mass index (kg/m$^2$).

We next adjusted the exposure–outcome association for the detected pleiotropic pathways and obtained an adjusted IVW estimate. The total heterogeneity, based on adjusting only these two of 157 SNP effects, was reduced by 17% ($Q = 567.6$). The effect estimate remained consistent with the original estimate, as did the IVW estimates when removing all outliers, or just outliers known to associate with candidate traits that associated with the outcome (Fig. 4a). However, the width of the confidence interval was substantially larger (including the null) after removing outliers known to associate with candidate traits (1 OR per SD: 1.80; 95% CI: 0.56, 5.79).

Example 2: Urate and coronary heart disease: Here we show an example with mixed findings from previous studies. The influence of circulating urate levels on risk of coronary heart disease has been under debate. Several MR studies have investigated the inflated effect of urate on CHD, which appeared to be influenced by pleiotropy[20,21]. We re-estimated the associations here using a range of MR methods. As has been previously reported the estimate from IVW suggested a weak association between urate and the risk of CHD using all variants (OR per 1 SD: 1.08; 95% CI: 1.00, 1.17), while there was a large intercept in the MR-Egger analysis (intercept = 1.02; 95% CI: 1.00, 1.03) with a much-attenuated causal effect estimate (Table 2). The median and mode-based estimates were also consistent with the MR-Egger estimate, indicating weak support for urate having a causal influence on CHD. Here, three variants were detected as outliers,

which associated with 61 candidate traits ($p < 5 \times 10^{-8}$). Among those outliers, rs653178 and rs642803 were associated with 14 traits that had conditionally independent influences on the outcome (Fig. 3b), including anthropometric measures (e.g. hip circumference), cholesterol levels, diagnosis of thyroid disease and smoking status.

Removing the outliers in the IVW analysis led to a more precise (though slightly attenuated) estimate of the influence of higher urate levels on CHD risk (OR per 1 SD: 1.05; 95% CI: 1.01, 1.10 and OR per 1 SD: 1.06; 95% CIs: 1.06, 1.12, respectively, Table 2). The adjustment model indicated an attenuated IVW estimate in comparison to the 'raw' approach, with confidence intervals spanning the null (OR per 1 SD: 1.07; 95% CI: 0.99, 1.16) while the degree of heterogeneity was reduced by half by accounting for the pleiotropic pathways through two outlier SNPs. The adjusted scatter plot showed that outliers moved towards the fitted line after controlling for the SNP effect on the candidate traits (Fig. 4b). The results in this analysis suggest that it is unlikely that urate has a strong causal influence on CHD. Here, outlier removal appears to strengthen evidence that may lead to a wrong conclusion.

Example 3: Sleep duration and schizophrenia: previous studies have shown that sleep disorder is associated with schizophrenia[22]. However, none of them confirmed the causality between sleep disorder and schizophrenia. We observed weak evidence for any association between sleep duration and schizophrenia (OR per 1

**Table 2 Results of empirical analyses with different IV estimators derived from various MR methods.**

| Methods | All variants | Estimates (95% CIs) | | |
|---|---|---|---|---|
| | | Removing outliers | Removing candidate outliers | Adjustment for candidate outliers |
| **Empirical analysis 1: systolic blood pressure (mmHg) on coronary heart disease (Odds ratio)** | | | | |
| Heterogeneity (Q)[a] | 682.7 (N SNPs = 157) | 312.1 (N SNPs = 150) | 448.7 (N SNPs = 155) | 567.6 (N SNPs = 157) |
| IVW random effects | 1.761 (1.474, 2.104) | 1.876 (1.655, 2.125) | 1.797 (0.558, 5.789) | 1.706 (1.449, 2.008) |
| Egger random effects | 2.641 (1.490, 4.679) | 2.951 (1.970, 4.419) | 2.206 (0.314, 15.472) | – |
| Intercept | 0.980 (0.969, 0.992) | 0.990 (0.982, 0.998) | 0.996 (0.988, 1.004) | – |
| Weighted median | 1.770 (1.528, 2.050) | 1.782 (1.539, 2.065) | 1.765 (0.576, 5.403) | – |
| Weighted mode | 1.770 (1.264, 2.479) | 1.726 (1.218, 2.447) | 1.740 (0.600, 5.043) | – |
| **Empirical analysis 2: urate (mg/dl) and coronary heart disease (Odds ratio)** | | | | |
| Heterogeneity (Q) | 81.6 (N SNPs = 24) | 20.7 (N SNPs = 21) | 33.4 (N SNPs = 22) | 44.1 (N SNPs = 24) |
| IVW random effects | 1.081 (0.996, 1.174) | 1.054 (1.008, 1.103) | 1.062 (1.057, 1.122) | 1.070 (0.992, 1.155) |
| Egger random effects | 0.952 (0.846, 1.071) | 1.008 (0.937, 1.084) | 0.990 (0.910, 1.077) | – |
| Intercept | 1.015 (1.003, 1.027) | 1.006 (0.998, 1.014) | 0.992 (0.984, 1.000) | – |
| Weighted median | 1.019 (0.961, 1.081) | 1.016 (0.958, 1.078) | 1.017 (0.961, 1.077) | – |
| Weighted mode | 1.028 (0.975, 1.084) | 1.022 (0.966, 1.082) | 1.025 (0.970, 1.083) | – |
| **Empirical analysis 3: sleep duration (hour/night) and schizophrenia (Odds ratio)** | | | | |
| Heterogeneity (Q) | 204.8 (N SNPs = 36) | 54.1 (N SNPs = 30) | 121.4 (N SNPs = 34) | 147.7 (N SNPs = 36) |
| IVW random effects | 1.184 (0.573, 2.445) | 1.289 (0.828, 2.008) | 1.215 (0.674, 2.192) | 1.181 (0.634, 2.197) |
| Egger random effects | 0.866 (0.056, 13.383) | 2.428 (0.485, 12.158) | 2.363 (0.254, 21.955) | – |
| Intercept | 1.004 (0.968, 1.042) | 0.991 (0.969, 1.013) | 0.991 (0.963, 1.020) | – |
| Weighted median | 1.276 (0.774, 2.104) | 1.249 (0.746, 2.090) | 1.250 (0.761, 2.052) | – |
| Weighted mode | 1.327 (0.679, 2.593) | 1.504 (0.728, 3.105) | 1.428 (0.702, 2.904) | – |
| **Empirical analysis 4: years of schooling (years) and body mass index (kg/m$^2$)** | | | | |
| Heterogeneity (Q) | 211.9 (N SNPs = 59) | 101.9 (N SNPs = 56) | 101.9 (N SNPs = 56) | 197.8 (N SNPs = 59) |
| IVW random effects | −0.272 (−0.386, −0.158) | −0.232 (−0.314, −0.150) | −0.232 (−0.314, −0.150) | −0.265 (−0.377, −0.153) |
| Egger random effects | 0.013 (−0.677, 0.703) | −0.404 (−0.910, 0.102) | −0.404 (−0.910, 0.102) | – |
| Intercept | −0.005 (−0.017, 0.007) | 0.003 (−0.005, 0.011) | 0.003 (−0.005, 0.011) | – |
| Weighted median | −0.209 (−0.307, −0.111) | −0.217 (−0.315, −0.119) | −0.217 (−0.315, −0.119) | – |
| Weighted mode | −0.141 (−0.413, 0.131) | −0.127 (−0.405, 0.151) | −0.127 (−0.405, 0.151) | – |

*N SNPs* number of single nucleotide polymorphisms, *95% CIs* 95% confidence intervals, *IVW* inverse variance weighted. Empirical analysis 1: systolic blood pressure (mmHg) and coronary heart disease (log odds); Empirical analysis 2: urate (mg/dl) and coronary heart disease (log odds); Empirical analysis 3: sleep duration (hour/night) and schizophrenia (log odds); Empirical analysis 4: years of schooling (years) and body mass index (kg/m$^2$).
[a]Heterogeneity amongst the estimates were assessed based on contribution of individual variant to Cochran's statistic.

SD: 1.18; 95% CIs: 0.57, 2.45), but there was substantial heterogeneity when all SNPs were used (Q = 204.8; p = 6.9 × 10$^{-26}$). Six outlier instruments were detected, which associated with 46 candidate traits (p < 5 × 10$^{-8}$). Among those outliers, the SNPs rs7764984 (near *HIST1H2BJ)* and rs13107325 (near *SLC39A8)* were associated with three traits that putatively influenced the outcome: self-reported coeliac disease, body composition (impedance of leg) and memory function (Fig. 4c).

We re-estimated the original association accounting for the detected outliers. The degree of heterogeneity was reduced by 74% (Q = 54.1) when removing all six outliers and by 46% (Q = 147.7) when adjusting for the two SNP effects that had putative pleiotropic pathways. Both methods of outlier removal and adjustment provide similar estimates in terms of direction, while the magnitude of estimates differed. After removing outliers, MR-Egger causal estimates were substantially larger (OR per 1 SD = 2.43; 95% CI: 0.49, 12.16 and OR per 1 SD = 2.36; 95% CI: 0.25, 21.96, respectively) than those from the method using all variants. IVW causal estimates from the adjustment method were virtually identical with the original estimates, with narrower CIs (OR per 1 SD = 1.18; 95% CI: 0.63, 2.20). While all methods indicate that sleep duration is unlikely to be a major causal risk factor for schizophrenia, pursuing outliers in the analysis provided putative indications that coeliac disease and memory function may be risk factors for schizophrenia (Fig. 4d).

Example 4: Years of schooling and body mass index: The association of education and health outcome is well established in social science[23]. Higher socioeconomic position is generally thought to lead to a lower risk of obesity in high-income countries[24,25]. We used 59 independent genetic instruments[26] to estimate the influence of years of schooling on BMI[27] (Table 2). All MR methods indicated that years of schooling has a causal beneficial effect on BMI (e.g. IVW beta: −0.27; 95% CI: −0.39, −0.16), except the estimate from MR Egger which had a very imprecise estimate (beta: 0.01; 95% CI: −0.67, 0.70), but the degree of heterogeneity was large (Q = 211.9 on 59 SNPs; p = 2.20 × 10$^{-8}$). Three outliers (rs6882046 near *LINC00461*, rs4800490 near *NPC1*, rs8049439 near *ATXN2L*) were identified as contributors to heterogeneity, and they showed associations (p < 5 × 10$^{-8}$) with 48 candidate traits. Among those candidate traits, two were associated with BMI (Fig. 3b): alcohol intake frequency (which associated with all three outliers) and usual walking pace.

We next re-estimated the influence of years of schooling on BMI by accounting for outliers. Adjusting the outliers for candidate trait pathways such as alcohol intake and usual walking pace reduced heterogeneity by 15% and had a small reduction in the confidence intervals while the point estimate remained consistent (Table 1). By contrast, there was a 48% reduction in heterogeneity when removing outliers. Point estimates remained largely consistent across all outlier removal methods. However, we note that Fig. 4b shows that one of the outliers (rs4800490, near gene *NPC1*) on the scatter plot moved away from the fitted line after adjusting for the pleiotropic pathway, indicating that if this outlier is due to a pleiotropic pathway we have estimated its indirect effect inaccurately or partially (e.g. where GWAS summary statistics are not available to identify other influential pleiotropic pathways).

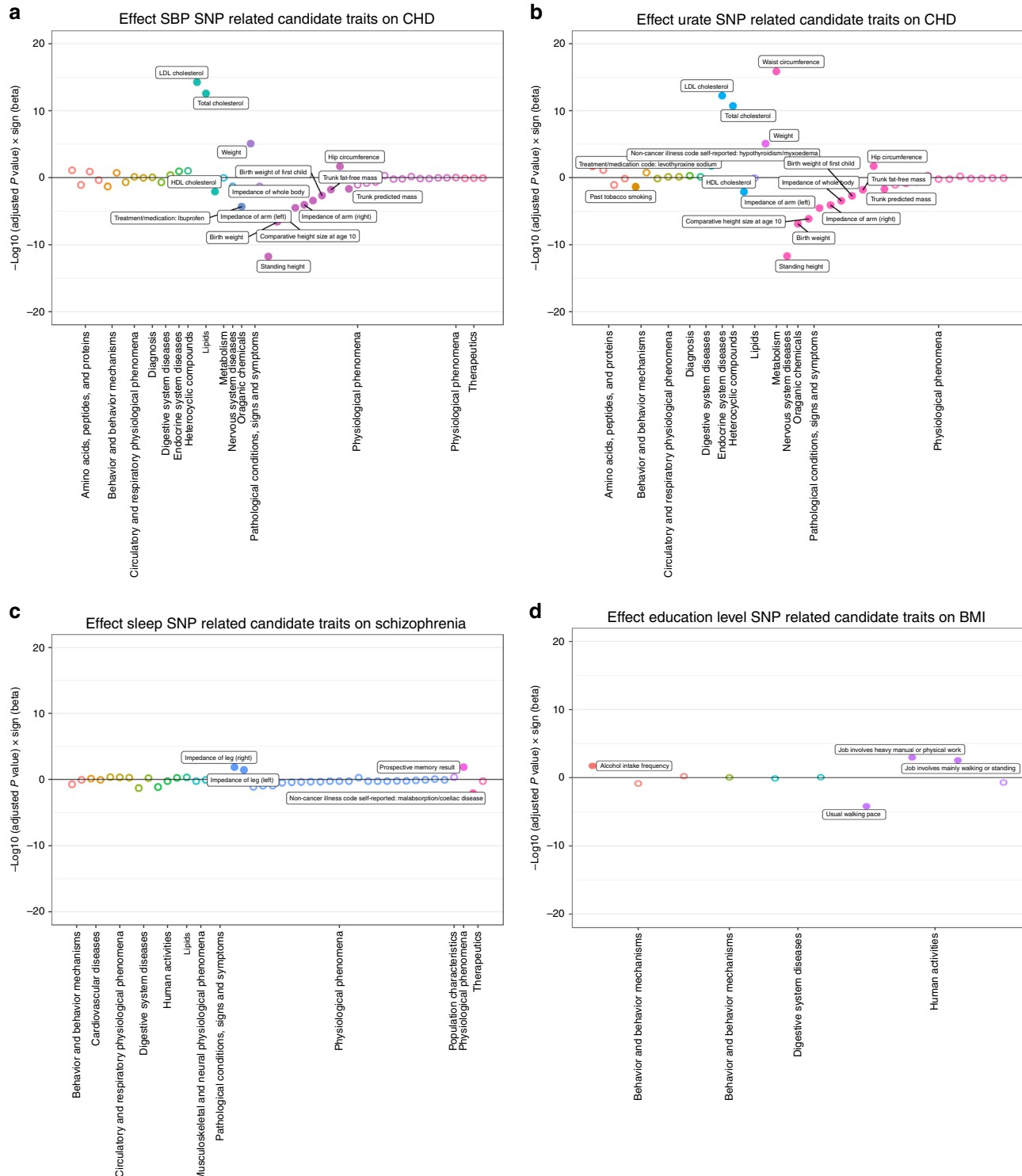

**Fig. 3 Causal associations between candidate exposures and hypothesised outcome.** Each candidate trait related to an outlier from an analysis is represented by a point in these plots. Along the x-axis, different phenotype groups are shown in different colours. The y-axis presents log transformed P value for each trait, multiplied by the sign of the causal effect estimate on the outcome. Filled circles in each category indicate the evidence of association between candidate traits and exposure or outcome (using an FDR < 0.05 threshold; see Methods for discussion of this). **a** Empirical analysis 1: systolic blood pressure (mmHg) and coronary heart disease (log odds). **b** Empirical analysis 2: urate (mg/dl) and coronary heart disease (log odds). **c** Empirical analysis 3: sleep duration (hour/night) and schizophrenia (log odds). **d** Empirical analysis 4: years of schooling (years) and body mass index (kg/m²).

## Discussion

The problem of instrumental variables being invalid due to horizontal pleiotropy has received much attention in MR analysis. Detecting and excluding such invalid instruments, based on

whether they appear to be outliers in the analysis, is now a common strategy that exists in various forms[7,8,14,15,28]. We have shown here that outlier removal could, in some circumstances, compound rather than reduce bias, and misses an opportunity to

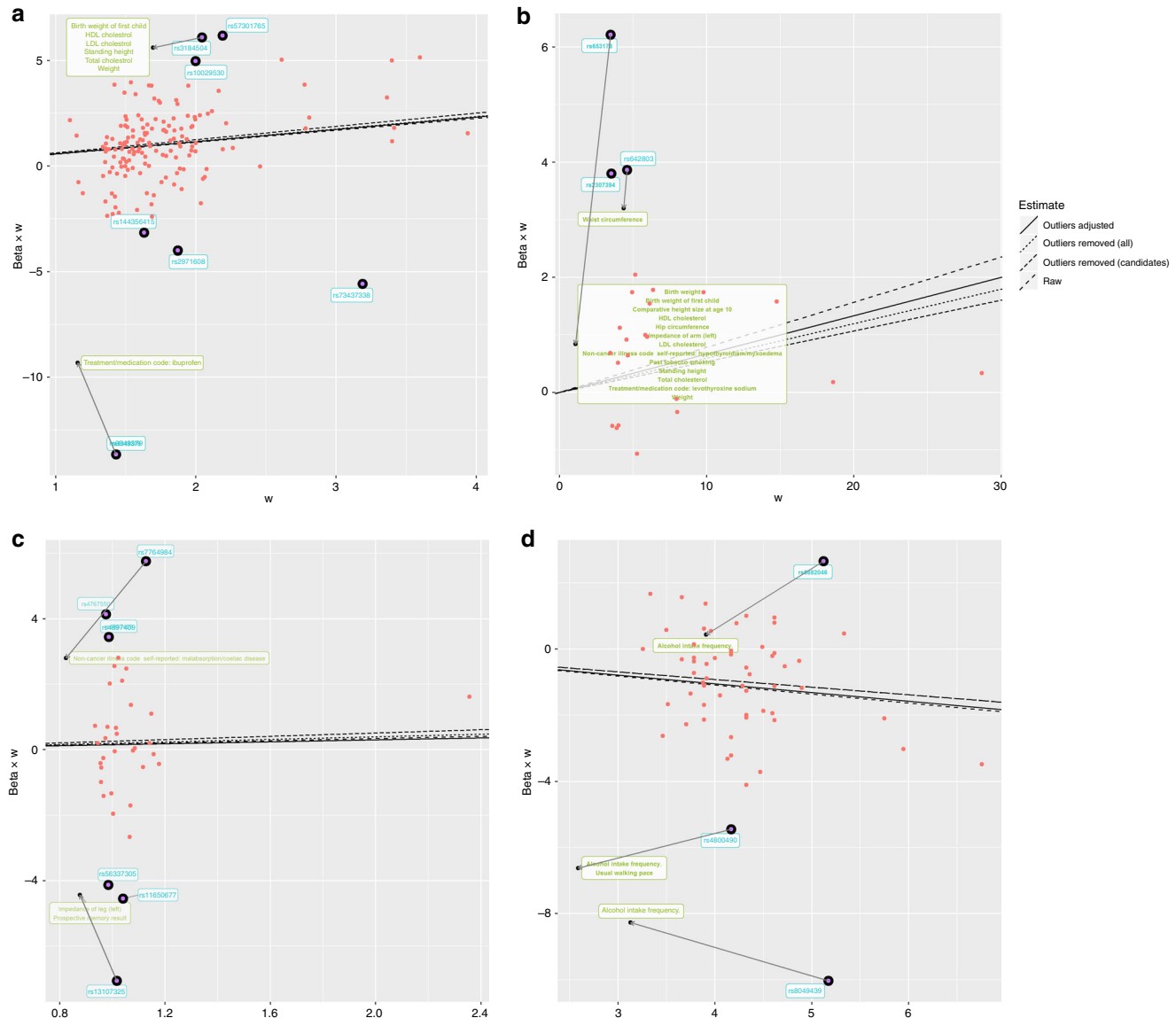

**Fig. 4 Exposure–outcome association adjusting the SNP effects on the candidate traits.** Radial plots of MR associations. The *x*-axis represents the weight (w) that each SNP contributes to the overall estimate, and the *y*-axis represents the product of the causal effect and weight of each SNP. The slopes represent causal effect estimates from different models (linetype). The arrows in this radial scatter plot indicates changes in the SNP's contribution to the overall causal effect estimate after conditioning on the effect of candidate traits on the outcome. The candidate traits that influence the association of the original exposure and the original outcome were listed in the box. **a** Empirical analysis 1: systolic blood pressure (mmHg) and coronary heart disease (log odds). **b** Empirical analysis 2: urate (mg/dl) and coronary heart disease (log odds). **c** Empirical analysis 3: sleep duration (hour/night) and schizophrenia (log odds). **d** Empirical analysis 4: years of schooling (years) and body mass index (kg/m²). Note that we use radial plots here as they explicitly show that one consequence of SNP-outcome effect adjustment is that the standard errors get larger (lower values on the *x*-axis). This leads to the adjusted variant contributing less weight to the causal effect and heterogeneity estimates, a process that acts in concert with the intention of attenuating the pleiotropic effect.

better understand the traits under study. We developed the MR-TRYX framework, which utilises the MR-Base database[10] of GWAS summary data to identify potential explanations for outlying SNP instruments, and to improve estimates by accounting for the pleiotropic pathways that give rise to them. We have also demonstrated the use and interpretation of MR-TRYX in four sets of empirical analyses.

To be effective, MR-TRYX depends upon the performance of three methodological components: (i) detecting instruments that exhibit horizontal pleiotropy; (ii) identifying the candidate traits on the alternative pathways from the variant to the outcome; and (iii) adequately estimating the effects of the candidate traits on the outcome. Each of these components present difficult problems,

but they are all modular and build upon existing methods and resources, and the MR-TRYX framework will naturally improve as those methods and resources themselves improve. We will now discuss the consequences of underperformance of each of these components on the TRYX analysis.

First it is important to notice that a major motivation for development of MR is that observational associations are often deemed unreliable because it is impossible to prove that there is no residual or unmeasured confounding biasing the effect estimate. But somewhat ironically, we find ourselves in a situation now where horizontal pleiotropy poses a similar challenge, in that proving that it is either absent or perfectly balanced is impossible. While several 'pleiotropy-robust' methods attempt to model out

pleiotropic effects by assuming a particular model of genetic architecture, another strategy is to adjust for horizontal pleiotropy, by including in the same model the genetic effects on one or more traits that are hypothesised to mediate the horizontal pleiotropic pathways (e.g. MVMR[29]). The adjustment approach depends upon those pathways being identified, which leaves it in a similar predicament to observational associations in that we cannot easily prove that all biasing pathways have been included in the model. The MR-TRYX approach falls within this category also, but we note that as fewer and fewer of the biasing pathways are identified and available to the adjustment model, the adjusted estimate will tend towards the IVW random effects estimate, which our simulations indicate can have good performance compared to, e.g., outlier removal methods. So, while clearly not a panacea for causal inference analysis, it is a valuable method within the MR toolkit, and its efficacy has been demonstrated. There is also an important contrast between outlier adjustment and multivariable MR in that the formulation of the latter is to estimate the direct effect of each exposure conditional on the others, whereas the former is to obtain an unbiased estimate of the total effect. MVMR may fail to distinguish between a pleiotropic model where the exposure (X) does not influence the outcome (Y) but has instruments that associate with another trait (A) which does influence Y, vs. a causal model in which trait A mediates the causal effect of X on Y. In both situations X will be deemed to be non-causal, despite it being indirectly causal in the latter case. This issue is discussed in detail elsewhere[19]. Here, outlier adjustment improves on the matter because MVMR will nullify all instruments for the exposure after adjusting for the mediator, leading to the exposure being dropped. When only the outlier variants are adjusted, the risk of erroneously removing the entire exposure signal is replaced by the lesser risk of incorrectly nullifying the effects of the outliers only. This will introduce heterogeneity and slight bias but is unlikely to remove the exposure's entire signal.

The classification of an outlier in MR analysis can be based on the statistical estimates of how a SNP being included as an instrument is due to being reverse causal (Steiger filtering)[12,17], the extent to which a single SNP disproportionately influences the overall result (e.g. Cook's distance), or most commonly the extent to which an SNP contributes to heterogeneity (e.g. Cochran's Q statistic, MR-PRESSO, and implicitly in median- and mode-based estimators)[7,8,14,15]. The philosophy of the latter two approaches is that proving horizontal pleiotropy is impossible, but that it should lead to outliers[9]. While a useful approximation, these approaches have two main limitations. First, determining whether a SNP is an outlier depends on the use of arbitrary thresholds, and this entails a trade-off between specificity and sensitivity. Second, if most variants are pleiotropic, then it is possible that the outlier SNPs are the valid instruments. Such a scenario can arise for complex traits such as gene expression or protein levels that have a few large effects and many small effects. For example, for C-reactive protein (CRP) levels, the SNP in the *CRP* gene region is likely the only valid instrument in some analyses[30]. In this context, bias due to horizontal pleiotropy cannot be avoided by selection of instruments since this approach may generate more bias[31]. This is supported by our simulation which demonstrates that in the presence of extensive pleiotropy removing outliers increased FDR and bias.

MR-TRYX should, in principle, avoid the problem of outlier removal because instead of removing outliers in their entirety, it attempts to eliminate the component of the SNP–outcome effect that is due to horizontal pleiotropy. Hence, we avoid implicitly cherry picking from among the SNPs to be used in the analysis, and if we have low sensitivity (i.e. a more relaxed threshold for outlier detection) it does not mean that there will be an unnecessary loss of power in the overall analysis. Previous work has

adjusted for the effect of pleiotropic phenotypes, but they treated pleiotropic phenotypes as exogenous variables that are not associated with the causal pathways of interest[32]. In MR-TRYX, candidate traits are treated as endogenous variables to account for the effect of the traits on the original association. Moreover, our method is applicable in the two-sample context, whereas the previous method requires individual level data. The problem of outlier detection which remains in MR-TRYX could be sidestepped by applying the adjustment approach to all SNPs irrespective of their contributions to heterogeneity.

Upon identification of potentially pleiotropic SNPs, MR-TRYX can only account for these if the pathways through which pleiotropy is acting can be identified. Detecting the pathways depends on the density and coverage of the human phenome available for the analysis. We use the MR-Base database of GWAS summary results, which comprises several hundred independent traits (we selected 605 traits from UK Biobank and 342 other complex traits and diseases obtained from previous GWA studies). While being the largest available resource, it is certainly not covering the whole human phenome. Therefore, even if a pleiotropic variant is detected correctly, it may not be possible to adjust it away if the phenotype associated with the variant cannot be identified. In the empirical analyses, often fewer than half of the candidate traits were inferred to be associated with the outcome. Yet, as we illustrated, MR-TRYX allows for an informative analysis that could routinely be applied in MR analyses. Broadening phenotype coverage is an on-going pursuit that will continually improve MR-TRYX analysis[33]. It is also important to note that in estimating the adjusted effect, the SNP–outcome standard error is liable to increase, which is one avenue through which heterogeneity is reduced as its outlying contribution will be down-weighted in the subsequent IVW analysis. We used radial MR plots to illustrate this explicitly in Fig. 4.

MR-TRYX is an automated framework, and this comes with several limitations in addition to those discussed already. First, our LASSO extension to multivariable MR is used to automate the selection of exposures that will be used for adjustment. A shrinkage step of LASSO may increase the SNP–exposure effect heterogeneity, which is necessary to assess the power of multivariable MR[34]. Multivariable MR is adept at establishing conditionally independent exposures but the reason that some exposures have attenuated effects in comparison to their total effects could be because (a) their total effects were biased by pleiotropy or (b) they are mediated by the exposures that are included in the model. Interpretations of (a) and (b) are very different, because in the case of mediation the exposure is a causal factor for the outcome. Second, we were primarily using the multivariable approach for practical purposes to avoid having multiple highly related exposures taken forward to the adjustment step (e.g. multiple different measures of body composition such as body weight and BMI). This approach worked effectively, although a problem remains unsolved in automating the removal of traits that are similar to the outcome. For example, if a trait similar to the outcome CHD associates with an outlier and is included in the multivariable analysis of multiple exposures against CHD, then all the other putative exposures will be dropped from the model. In the analyses presented we manually removed traits that came up as candidate pleiotropic pathways but were, in fact, synonymous with or closely related to the outcome. Third, we note that heterogeneity does not necessarily arise only because of pleiotropy, for example the non-collapsibility of odds ratios will introduce heterogeneity automatically which cannot be adjusted away through the TRYX approach. Many other mechanisms exist that can lead to bias in MR, as has been described in detail elsewhere. Fourth, SNPs can appear to be outliers not through being pleiotropic, but through

other mechanisms, such as population stratification (association of alleles with phenotypes being confounded by ancestral population), canalisation (developmental compensation to a genetic change)[2,35], or the influence on phenotype being changeable across the life course[36]. Fifth, since MR-TRYX uses the resource from MR-Base, it is recommended that the user acknowledge the limitation and restriction of MR-Base[10]. For example, the population should be the same for the exposure (or the candidate traits) and the outcome traits to avoid mis-estimation of the magnitude of the association. Also, sample overlap should be recognised between the GWAS studies for the SNP–exposure and SNP–outcome association to prevent effect estimates being biased[37]. Users should consider modifying their analyses when the limitations indicated above are avoidable. Sixth, in the case of a binary outcome, there may be parametric restrictions on the conditional causal odds ratio in our multivariable MR model where the exposure effect is linear in the exposure on the log odds ratio scale[38]. However, the two-stage estimator with a logistic second-stage model still yields a valid test of the causal null hypothesis[38]. Finally, it is necessary for the effects through the identified pleiotropic pathways to be accurately estimated. This is a recursive problem—MR-TRYX adjusts the SNP–outcome effects based on the pleiotropic effect through the outlier SNP, but it does this by introducing more SNPs into the analysis that instrument the candidate traits. These new SNPs may themselves exhibit pleiotropic effects that could lead to bias in the estimates of the candidate traits on the outcome, requiring a second round of TRYX-style candidate trait searches, and so on. In the example of education level and BMI, adjustment for the pleiotropic pathway failed to substantially reduce the degree of heterogeneity. Further developments could involve recursively analysing alternative pathways. For example, Steiger filtering could be applied at all stages of MR estimation to attempt to automatically remove reverse causal instruments or those that arise due to confounding pleiotropy[17].

In this study, we demonstrated the use of MR-TRYX through four examples of identifying putative pathways. In the first empirical example (SBP on CHD), we illustrated the validity of MR-TRYX to detect the traits that possibly influence the disease outcome. Apart from SBP, MR-TRYX also detected well-established risk factors for CHD including adiposity, cholesterol levels, and standing height. An interesting finding from this example is that headache-related traits (e.g. experience of pain due to headache and self-reported status of ibuprofen intake) were identified as candidate traits, which may influence the original association. In support of the putative finding for self-reported ibuprofen use associating with CHD, we also found that pain experienced in the last month (headache) and self-reported migraine were associated with lower risk of CHD (OR per 1 SD: 0.33; 95% CI: 0.12, 0.89 and beta = 0.02; 95% CI: 0.0004, 0.65, respectively). A previous study reported shared genetic risk between headache (migraine) and CHD, suggesting a potential role of migraine in vascular mechanisms[39]. An alternative mechanism that could give rise to this association is that the effect of pain on lower CHD risk is mediated through the use of medications such as aspirin that have known protective effects on CHD.

The example of urate and CHD demonstrated the benefit of the adjustment method showing that the noise due to pleiotropy was substantially reduced after correcting for the effect of candidate traits. The presence of hypothyroidism and self-reported levo-thyroxine sodium intake status were identified as putative risk factors for risk of CHD, which is consistent with previous clinical trials: thyroid dysfunction is associated with overall coronary risk[40], which can be reversed by levothyroxine therapy[41]. In the education–BMI example, we showed that increased alcohol intake

and slower usual walking pace may influence obesity. These identified traits have been reported as possible risk factors for higher BMI and obesity[42,43]. Additionally, the example of sleep duration and risk of schizophrenia suggested coeliac disease and body composition as putative risk factors for schizophrenia. A number of observational studies suggested that schizophrenia is linked with body composition[44] and coeliac disease[45]. MR of binary exposures is often difficult to interpret because the instrument effects are on liability to disease, not the presence or absence of the disease. Hence, the association between coeliac disease and schizophrenia may be better interpreted as an indication of shared disease aetiology. Nevertheless, this is a valuable finding since the causal effect of those putative risk factors on risk of schizophrenia has not been investigated using an MR approach. Therefore, our example illustrates how outliers can be used to identify alternative pathways, opening the door for hypothesis-free MR approaches and a network-based approach to disease.

In conclusion, we have introduced a framework to deal with the bias from horizontal pleiotropy, and to identify putative risk factors for outcomes in a more directed manner than typical hypothesis-free analyses, by exploiting outliers. Heterogeneity is widespread across MR analyses and so we are tapping into a potential new reservoir of information for understanding the aetiology of disease. The strategy is a departure from previous ones dealing with pleiotropy—enlarging the problem by searching across all traits for a better understanding of a specific exposure–outcome hypothesis can be fruitful.

## Methods

**Outlier detection.** Several outlier detection methods now exist that are based on the contribution of each SNP to overall heterogeneity in an IVW meta-analysis[46]. In order to estimate heterogeneity accurately, it is important to appropriately weight the contribution of each SNP to the overall estimate. We used the approach implemented in the RadialMR R package (https://github.com/WSpiller/RadialMR) to detect outliers. Full details are provided elsewhere[15], but briefly, we used the so-called 'modified 2nd order weighting' approach to estimate total Cochran's $Q$ statistic as a measure of heterogeneity, as well as the individual contributions of each SNP, $q_i$[15]. This has been shown to be comparable to the simulation-based approach in MR-PRESSO, providing a well-calibrated test statistic for outlier status whilst being computationally more efficient[14,47]. The probability of a SNP being an outlier is calculated based on $q_i$ being chi-square distributed with one degree of freedom. For demonstration purposes we adopted a $p$ value threshold that was Bonferroni corrected for the number of SNPs tested in analysis ($p < 0.05$/number of SNPs). We are not, however, suggesting that this arbitrary threshold will necessarily be optimal for identifying outliers, and users can apply other approaches or thresholds through the MR-TRYX software.

**Candidate trait detection.** Traits associated with the detected outliers could causally influence the outcome. MR-TRYX searches the MR-Base database to identify the traits that have associations with the detected outliers. By default, we limit the search to traits for which the GWAS results registered at MR-Base have more than 500,000 SNPs and sample sizes exceeding 5000. Traits that have an association with outlier SNPs at genome-wide $p$ value threshold ($p < 5 \times 10^{-8}$; in keeping with traditional GWAS thresholds used for instrument selection) are regarded as potential risk factors for the outcome and defined as candidate traits. Each candidate trait is tested for its influence on the original exposure (X) and outcome (Y) traits (Fig. 1) using the IVW random effects model. We take forward putative associations based on false discovery rate (FDR) < 0.05, where the null hypothesis is true, but we note that the use of arbitrary thresholds is problematic[48,49], and we use them here to make high dimensional investigations more manageable.

**Assessing effect of the candidate traits on the outcome.** Once candidate traits are detected, we can identify instruments specifically for the candidate traits and model how the exposure and candidate traits together associate with the outcome. This involves the following process, which we go on to describe in full detail below:

1. Identify instruments for the candidate traits.
2. Estimate the influence of the candidate traits on $y$ conditioning on $x$ using multivariable MR.

Suppose we have $g_0, g_{x1}, \ldots, g_{xE}$ instruments for the exposure $x$ where $g_0$ is an outlier in the $x$–$y$ MR analysis due to an association with candidate trait $P$,

and where $E$ indicates the number of genetic variants for the exposure. Also, $P$ has $g_0, g_{P1}, \ldots, g_{PM}$ genetic instruments, where $M$ is the number of genetic variants for $P$. To obtain the estimate of ($py$) uncontaminated by shared genetic effects between $P$ and $x$ (Fig. 1a), we perform multivariable MR analysis[34]. We generate a combined list of instruments for both $x$ and $P$ and clump them to obtain a set of independent SNPs. The original outlier is removed from amongst these SNPs. We then obtain the genetic effects of each of these SNPs on the exposure ($gx$), candidate trait ($gp$), and outcome ($gy$). Finally, we estimate the causal influence of $P$ on $y$ conditioning on $x$ by regressing ($gy$) ~ ($gx$) + ($gp$) weighted by the inverse of the variance of the ($gy$) estimates. The whole process is automated within the TwoSampleMR R package which connects to the MR-Base database.

In the case of an outlier SNP associating with many candidate traits we first apply a modified form of multivariable MR, involving LASSO regression of ($gy$) ~ ($gx$) + ($gp_i$) + ... + ($gp_p$) and use cross-validation to obtain the shrinkage parameter that minimises the mean squared error. We retain only the candidate traits that are putatively associated with the outcome and have non-zero effects after shrinkage. Then we apply remaining traits in a multivariable model with $x$ against the outcome, as described above[34]. We perform the LASSO step because many traits in the MR-Base database have considerable overlap and redundancy, and the statistical power of multivariable analysis depends on the heterogeneity between the genetic effects on the exposure variables[34]. Using LASSO therefore automates the removal of redundant traits (Supplementary Fig. 1, Supplementary Tables 2 and 3). We then obtain estimates of ($py$) that are conditionally independent of $x$ and jointly estimated using all remaining $P$ traits by combining them in a multivariable analysis on the outcome $y$. A detailed discussion of dealing with multiple candidate traits per outlier SNP is presented in Supplementary Note 1.

**Adjusting causal estimates for candidate-trait associations.** An illustration of how outliers arise in MR analyses is shown in Fig. 1. If a SNP $g$ has some influence on exposure $x$, and $x$ has some influence on outcome $y$, the SNP effect on $y$ is expected to be ($gy$) = ($gx$)($xy$), where ($gx$) is the SNP effect on $x$ and ($xy$) is the causal effect of $x$ on $y$. Any substantive difference between ($gy$) and ($gx$)($xy$) could be due to an additional influence on $y$ arising from the SNP's effect through an alternative pathway.

If a SNP influences a 'candidate trait', $P$, which in turn influences the outcome (or the exposure and the outcome), then the SNP's influence on the exposure and the outcome will be a combination of its direct effects through $x$ and indirect effects through $P$[34]. If we have estimates of how the candidate trait influences the outcome, then we can adjust the original SNP–outcome estimate to the effect that it would have exhibited had it not been influencing the candidate trait. In other words, we can obtain an adjusted SNP–outcome effect conditional on the 'candidate-trait–exposure' and 'candidate-trait–outcome' effects. If the SNP influences $P$ independent candidate traits (as selected from the LASSO step), then the expected effect of the SNP on $y$ is

$$(gy) = (gx)\widehat{(xy)} + \sum_{i=1}^{P}(gp_i)\widehat{(p_iy)}. \tag{1}$$

Hence, the effect of the SNP on the outcome adjusted for alternative pathways $p_1, \ldots, p_P$ is

$$(gy)^* = \widehat{(gy)} - \sum_{i=1}^{P}(gp_i)(p_iy). \tag{2}$$

We use parametric bootstraps to estimate the standard error of the $(gy)^*$ estimate, where 1000 resamples of ($gy$), ($gp$), and ($py$) are obtained based on their respective standard errors and the standard deviation of the resultant $\widehat{(gy)}^*$ estimate represents its standard error. Finally, an adjusted effect estimate of ($xy$) due to SNP $g$ is obtained through the Wald ratio.

$$\widehat{(xy)} = \frac{(gy)^*}{(gx)}. \tag{3}$$

Occasionally it might be possible that a candidate trait $P$ is a redundant trait for $y$, for example if the outcome is coronary heart disease, the outliers might detect traits such as 'medication for heart disease' as a potential candidate trait. It would make no sense to attempt to adjust the SNP–outcome association for a trait that is essentially the same as the outcome, it would just nullify the association. We have not yet developed an automated method to remove such traits, but we recommend manually checking any traits that are selected for automated outlier adjustment.

**Simulations.** IVW effect estimates are liable to be biased when at least some of the instrumenting SNPs exhibit horizontal pleiotropy, and those SNPs tend to contribute disproportionately towards the heterogeneity in the effect estimate. We conducted simulations to evaluate how different methods perform at estimating the causal effect of $x$ on $y$ under different circumstances. The simulations are principally designed to evaluate the potential value of adjusting outliers for putative explanatory pathways. Other aspects of the MR-TRYX framework, for example, dealing with redundant traits in the GWAS database are dealt with separately (Supplementary Note 1). In all circumstances there are 30 independent genetic effects on $x$ ($Gx$), and $x$ either has no direct influence on $y$, or has a direct effect of 0.1 on $y$. For all simulations, we used 10,000 individuals, and repeated each

circumstance 1000 times. We summarised each scenario in two ways: (a) We estimated the proportion of simulations that gave a biased estimate of the causal effect of $x$ on $y$ ($b_{xy}$). For each simulation we calculated the probability of the effect estimate being substantially different from the true simulated effect based on whether the true effect fell outside the 95% confidence interval of the estimate. Then for the set of 1000 simulations, we calculated the proportion of estimates that were 'unbiased'. (b) We summarised the power and FDR by estimating the area under the receiver operator curve, characterising the sensitivity and specificity of each method at determining whether the true causal effect estimate was null or non-null. Each simulation is conducted by first simulating data to satisfy the parameters described below. We then search for instruments for $x$ across all simulated genetic variants and retain those that are significant after Bonferroni correction, and applying the summary data-based methods based on the genetic associations for the instruments on $x$ and $y$. All genetic variants are simulated to be Hardy Weinberg equilibrium with an allele frequency of 0.5.

We investigated three scenarios that could give rise to invalid instruments (Fig. 2).

In the confounding pleiotropy scenario, there are instruments detected for $x$ that primarily influence a confounder variable (e.g. $u_1$ that influences both $x$ and $y$). Therefore, the term 'confounding pleiotropy' indicates that the instrument's horizontal pleiotropic effect arises because it primarily influences a confounder of $x$ and $y$. See Fig. 2 (column 1) for a DAG describing the model. The confounder $u_1$ has a set of independent genetic influences, $G_{u1}$, which may be detected as instruments for $x$.

$$u_1 = \sum_{j}^{m_{u1}} G_{u1,j} b_{gu1,j} + e_{u1}, \tag{4}$$

$$x = u_1 b_{u1x} + \sum_{j}^{m_x} G_{x,j} b_{gx,j} + e_x, \tag{5}$$

$$y = u_1 b_{u1y} + x b_{xy} + e_y. \tag{6}$$

Parameters: $b_{gu1,j}$ values are sampled for each SNP $m_{u1}$ from a normal distribution such that they explain 60% of the variance in $u_1$. The value of $b_{u1x}$ is chosen such that $u_1$ explains 60% of the variance in $x$ and 40% of the variance in $y$. The values of $b_{gx,j}$ are sampled from a normal distribution for each of $m_x$ SNPs such that they explain 20% of the variance in $x$. The causal effect $b_{xy}$ is set to either 0, or some value such that $x$ explains 10% of the variance in $y$. Values for $e_{u1}, e_x$, and $e_y$ are sampled from normal distributions with mean 0 and variances that are scaled to satisfy the variances of all other parameters described for the model. Different sets of simulations are run with different proportions of invalid instruments by simulating different numbers of genetic variants directly influencing $u_1$ or $x$:

$$m_{u1} \in \{5, 10, 15, 20, 30\};$$

$$m_x \in \{5, 10, 15, 20, 25, 30\}.$$

In the case of horizontal pleiotropy, at least some of the instruments for $x$ have an independent effect on $y$ that is mediated through some other pathway that does not include $x$. In these simulations, the pleiotropic influence of each instrument, $G_{x,i}$, is mediated by a different trait, $u_{2,i}$

$$x = \sum_{j}^{m_x} G_{x,j} b_{gx,j} + e_x, \tag{7}$$

$$u_{2,i} = \sum_{j}^{m_{u2,i}} G_{u2,i,j} b_{gu2,i,j} + G_{x,i} b_{plei,i} + e_{u2,i}, \tag{8}$$

$$y = \sum_{i}^{m_{plei}} u_{2,i} b_{u2,i,y} + x b_{xy} + e_y. \tag{9}$$

Parameters: Some number $m_{plei} \in \{5, 10, 15, 20, 25, 30\}$ of 30 $G_x$ instruments for $x$ are selected to have pleiotropic effects, such that they influence $y$ each mediated by an independent trait $u_{2,i}$ which itself has its own set of 30 direct genetic influences $G_{u2},i$. The $b_{gu2,i,j}$ values for the genetic effects on $u_{2,i}$ are sampled from a normal distribution such that they explain 20% of the variance in $u_{2,i}$. Each pleiotropic $G_{x,i}$ instrument has an influence on $u_{2,i}$ that explains 20% of its variance ($b_{plei,i}$). The influence of each $u_{2,i}$ on $y$ is such that $b_{u2,i,y}$ is normally distributed with mean 0 and variance 0.4. The outcome $y$ is also influenced by $x$ where the causal effect $b_{xy}$ is set to either 0, or some value such that $x$ explains 10% of the variance in $y$. Values for $e_{u2,i}, e_x$, and $e_y$ are sampled from normal distributions with mean 0 and variances that are scaled to satisfy the variances of all other parameters described for the model.

Mediation pleiotropy is treated as in 'confounding pleiotropy', except the pleiotropic relationships arise due to a trait that is mediating the path from $x$ to $y$, rather than confounding it (Fig. 2, column 3). Specifically, the influence of $x$ on $y$ is at least partially mediated by another trait $u_3$, and at least some of the instruments

for $x$ have an independent pleiotropic influence on $u_3$.

$$x = \sum_{j}^{m_x} G_{x,j} b_{gx,j} + e_x, \qquad (10)$$

$$u_3 = \sum_{j}^{m_{u3}} G_{u3} b_{gu3,j} + \sum_{i}^{m_{plei}} G_{x,i} b_{plei,i} + x b_{x,u3} + e_{u3}, \qquad (11)$$

$$y = u_3 b_{u3,y} + x b_{xy} + e_y. \qquad (12)$$

Parameters: Some number $m_{plei} \in \{5, 10, 15, 20, 25, 30\}$ of 30 $Gx$ instruments for $x$ are selected to have pleiotropic effects, such that they influence $u_3$ which itself mediates an effect from $x$ to $y$, and has its own set of 30 direct genetic influences $G_{u3}$. The $b_{gu3,j}$ values for the genetic effects on $u_3$ are sampled from a normal distribution such that they explain 20% of the variance in $u_3$. Each pleiotropic $Gx,i$ instrument has an influence on $u_3$ such that $b_{plei,i}$ are sampled from a normal distribution explaining 20% of the variance in $u3$ in total. The indirect influence of $x$ on $y$ is generated such that $x$ explains 30% of the variance in $u_3$, and $u_3$ explains 40% of the variance of $y$. The outcome $y$ may also be influenced directly by $x$ where the causal effect $b_{xy}$ is set to either 0, or some value such that $x$ explains 10% of the variance in $y$. Values for $e_{u3}$, $e_x$, and $e_y$ are sampled from normal distributions with mean 0 and variances that are scaled to satisfy the variances of all other parameters described for the model.

In these simulations we ask: if we can identify the pathway through which an outlier SNP has a horizontal pleiotropic effect, can adjustment for that pathway improve the original exposure–outcome analysis? We assess the performance of the following methods for each simulation.

(1) Raw, where all detected instruments are used in a standard IVW random effects analysis.
(2) Adjusted SNP–outcome effects
　(a) Where outlier SNPs are tested for association with all candidate traits and adjusted for the effect of the candidate trait on the outcome using MR-TRYX.
　(b) Where attempts are made to adjust all detected instruments regardless of outlier status.
(3) Removed instruments
　(a) Where all detected outliers are removed.
　(b) Where only outliers that are found to influence a candidate trait are removed.
(4) Multivariable MR (MVMR)
　(a) Where the traits selected to be included in the model are the candidate traits associated with outliers.
　(b) Where the traits selected to be included in the model are the candidate traits associated with any of the detected instruments regardless of outlier status.

**Empirical analyses**. As applied examples, we chose two robust findings and two controversial findings that are potentially biased due to pleiotropy: (i) systolic blood pressure (SBP) and coronary heart disease (CHD); (ii) urate and CHD; (iii) sleep duration and schizophrenia; and (iv) education level (years of schooling) and body mass index (BMI). Those examples were chosen based on previous findings[20,22,50,51] to illustrate how pleiotropic variants can be used to identify other pathways and adjusted to estimate the causal effect of the original exposure on the outcome independent of pleiotropic bias.

Summary statistics (β-coefficients and SEs) for the associations of the SNPs with each exposure were obtained from the publicly available GWAS database (Supplementary Table 1). Selected SNPs were harmonised for the analysis, excluding palindromic SNPs and pruning for linkage disequilibrium ($r^2 < 0.001$). We primarily used the two-sample MR IVW method to obtain causal estimates between exposures and outcomes allowing each SNP to have a different mean effect (random effects model). A number of sensitivity analyses were applied to evaluate the consistency of causal effect estimates under different models of pleiotropy among the SNPs, including the MR-Egger[6], weighted median, and weighted mode approaches[7,8].

Outliers were detected among the instruments for each exposure ($p < 0.05/$the number of SNPs). We searched the MR-Base database to identify the candidate traits that are associated with outliers ($p < 5 \times 10^{-8}$). We then performed multivariable MR analysis to test which candidate trait can explain the heterogeneity in the original exposure–outcome association. To perform multivariable MR, more SNPs that instrument the candidate traits were introduced into the analysis.

Subsequently we re-estimated the association of the original exposure and the original outcome using different sets of instruments: (a) all SNPs (corresponding to the raw method in our simulation), (b) outliers adjusted, (c) all outlier removed, and (c) candidate outliers removed.

All analyses were conducted with the TwoSampleMR package (https://github.com/MRCIEU/TwoSampleMR) and the MR-TRYX package (https://github.com/explodecomputer/tryx) in R statistical software (ver 3.4.1). Detailed information are provided in Supplementary Note 1 and the scripts used for the simulations and empirical analyses can be found here https://github.com/explodecomputer/tryx-analysis.

**Reporting summary**. Further information on research design is available in the Nature Research Reporting Summary linked to this article.

## Data availability
The data that support the findings of this study are available from IEU GWAS database (https://gwas.mrcieu.ac.uk/).

## Code availability
A copy of the code used in this analysis is available at https://github.com/explodecomputer/tryx and https://github.com/explodecomputer/tryx-analysis.

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

## Acknowledgements

This study was supported by the UK Medical Research Council (MC_UU_00011/1; MC_UU_00011/4), which founds Integrative Epidemiology Unit at the University of Bristol where Y.C., P.C.H., E.S., T.R.G., J.Z., G.D.S. and G.H. work. G.H. was supported by the Wellcome Trust and Royal Society [208806/Z/17/Z].

## Author contributions

Y.C., G.D.S. and G.H. conceived the study and developed the statistical analysis plan. Y.C. and G.H. developed the model and methods. Y.C., G.D.S. and G.H. prepared the first draft of manuscript. Y.C., P.C.H., E.S., T.R.G., J.Z., A.P.M., G.D.S., and G.H. contributed to the writing of the manuscript. All authors reviewed and agreed on the manuscript.

## Competing interests

The authors declare no competing interests.
