## [Peer Review File · Nature Communications]

Reviewers' Comments:

Reviewer #1:

Remarks to the Author:

MR-TRYX: Exploiting horizontal pleiotropy to infer novel causal pathways

Overall impression

This paper combines several state-of-the-art statistical approaches from the Mendelian Randomization (MR) literature (e.g., outlier detection, LASSO, Multivariable MR) to (i) detect possible pleiotropic SNPs, (ii) infer novel causal pathways to the outcome of interest; and (iii) adjust the traditional IVW estimate for pleiotropic effects.

There is much to like about this paper: in particular, I find it refreshing that the authors discuss horizontal pleiotropy as an opportunity to detect novel causal pathways rather than just a nuisance to MR. I'm therefore convinced by parts (i) and (ii) of their analysis. Especially the part on exploring novel causal pathways is very interesting, doesn't require a lot of assumptions, and is user-friendly. I'm somewhat lukewarm on part (iii) – as I will elaborate upon below – but that part also expands the methodological toolbox of MR and enables researchers to triangulate across an even wider range of possible MR approaches.

Comments

- My main comment is on how reliable the last part of the MR-TRYX approach – adjusting the MR estimates – is. In particular, it relies on the strong assumption that you can obtain an unbiased estimate of all of the relevant candidate traits P on the outcome Y . To their credit, on page 12 and in the discussion section the authors discuss many caveats and are upfront about the potential limitations. However, I feel the authors could be even more prudent and present the last part of the analysis as a way to tentatively gauge the direction of bias in the original MR estimate rather than presenting it as a bias-corrected MR estimate.

MR-base is an extremely rich resource, and is expanding, but the MR-TRYX approach relies heavily on candidate traits being measurable and for which GWAS results exist. Therefore I feel a bit like you're back in the world of observational studies with unmeasured confounding: you're never sure whether you've adjusted for all the relevant candidate traits that bias the effect of X on Y .

One idea that came to mind is whether you could use negative controls or MRGxE to have some sense whether the adjustment through observable candidate traits is sufficient to eliminate bias. I mean, one way to purge the effect of X on Y from possible pleiotropic effects is to measure the relevant pleiotropic effects through observed candidate traits as the authors suggest. An alternative approach is to purge the effect of X on Y from pleiotropic effects using no-relevance groups for whom $X=0$ by definition as in Chen et al. (2008), Van Kippersluis & Rietveld (2018), and Spiller et al. (2019). If both methods suggest a similar correction of the original IVW estimate, this raises confidence in the correction part of MR-TRYX.

Chen L, Davey Smith G, Harbord R, Lewis SJ. Alcohol and blood pressure: A systematic review implementing a Mendelian Randomization approach. *PLoS Med* 2008;5:e52.

Hans van Kippersluis, Cornelius A Rietveld; Pleiotropy-robust Mendelian randomization, *International Journal of Epidemiology*, Volume 47, Issue 4, 1 August 2018, Pages 1279–1288,

Wes Spiller, David Slichter, Jack Bowden, George Davey Smith; Detecting and correcting for bias in Mendelian randomization analyses using Gene-by-Environment interactions, *International Journal of Epidemiology*, , dyy204, <https://doi.org/10.1093/ije/dyy204>

- My second main comment is that I believe the authors should talk more about the hypothesized data generating process under which their approach works. To me it seems that the approach works best when one has an additive model in which X and all the candidate traits P have additive effects, and where none of the candidate traits P is a mediator or collider in the relationship between X and Y. But what if P is a mediator of the relationship between X and Y? What if P is a collider? What if there are interaction effects between X and P?

Related, redundancy of candidate traits is currently determined by a statistical approach (LASSO) but shouldn't redundancy not also be based on some theory or at least some idea of how the candidate trait relates to X and Y in e.g., a Directed Acyclical Graph (DAG)?

- I think the description of multivariable MR is a bit confusing. In the introduction it is mentioned that the causal influence of the candidate trait on the outcome is obtained using MR excluding the original outlier; then on page 7 the authors talk about a set of T clumped instruments for both X and P; and then finally on page 10 it is mentioned that additional SNPs are introduced to instrument the candidate traits. Please fill me in on how exactly the multivariable MR is performed.
- Two comments on the selection of candidate traits and the differential use of false discovery rates:
 - Candidate traits are defined as those having an association with the outlier SNP at genome-wide significance level. However, since the p-value in the GWAS is a function of both the effect size but also the sample size, this way of selecting traits is partly on basis of the accidental sample size of the discovery GWAS and not on basis of the potential influence on the MR estimate under study. Would it not be better to select traits on basis of the effect size (and hence the potential influence on the MR estimate)?
 - Then candidate traits are further reduced by looking at the effect of these traits on the outcome. Here a p-value of 0.05 is used. In the outlier detection part a p-value of 0.05 divided by the number of SNPs is used as a correction for multiple testing. For consistency, wouldn't it be better to apply a similar correction to the 0.05 FDR here too?
- I don't find Figure 2 very illuminating. Could you also present the actual estimates in a Table such that readers can judge the bias? How is "substantially different" on line 665 defined?

Minor comments

- On page 1, line 56, you talk about 'outliers', but at this stage in the introduction it is not clearly defined what you mean by outliers. Perhaps you can help the reader by defining outliers as 'suspicious SNPs' or 'SNPs exhibiting possible pleiotropic effects'?

- Page 4, lines 111-113, this part is very hard to understand for non-experts. Could you provide a bit more intuition or at least a good reference as you did before on line 102.
- In empirical example 1, isn't ibuprofen use a potential collider?
- To fairly compare the confidence intervals across methods, and given that 1000 bootstrap replications is fair but not very large, would it be useful to also bootstrap the standard errors of the IVW approach under the same number of replications, to make sure that the differences in confidence intervals are due to the method of estimation and not the method of computing standard errors?
- I don't understand the sentence on line 434-435 on the shrinkage step.
- The legend and labels of the figures could be improved.

Reviewer #2:

Remarks to the Author:

This paper presents a comprehensive approach to investigate horizontal pleiotropy in Mendelian randomization (MR) studies. MR studies can provide useful evidence and complement well evidence from observational studies. The authors have not developed a new method of detecting and/or treating horizontal pleiotropy, but they have put together an algorithm and a nice compilation of steps to study horizontal pleiotropy using several existing databases and methods. This paper is a welcomed addition for the MR community, and provides further ways to investigate horizontal pleiotropy. The authors acknowledge that pleiotropy is ubiquitous in the genome and in MR studies and that their method is far from perfect, but they have developed an intelligent and balanced report. This method will improve as public availability of GWAS results improves. Overall, this work is useful for the field. I have a few relatively minor comments on the specifics of the paper:

- 1) The authors selected a P-value threshold of 5×10^{-8} for selecting SNPs exhibiting horizontal pleiotropy in associations with secondary traits and outcomes, which may be too conservative especially given GWAS with smaller sample sizes.
- 2) It would be useful if the authors could add the number of genetic instruments in the Results for the four practical examples that they use.

Reviewer #3:

Remarks to the Author:

Cho et al. propose a new statistical framework to investigate horizontal pleiotropy in the context of Mendelian randomization (MR) in order to discover novel causal pathways. Indeed, in MR, some instruments used to explore the causal link between a given exposure and an outcome are submitted to horizontal pleiotropy, meaning the instruments are associated with other unknown exposures causal to the outcome. Therefore, investigating association of these instruments submitted to horizontal pleiotropy in order to point out these unknown causal exposures is the goal of MR-TRYX. The authors employ a 4-step approach: 1) detection of the outlier instruments putatively submitted to horizontal pleiotropy; 2) Look for association between the outlier variants and traits in MR-Base database; 3) LASSO-based selection of these new candidate exposures; 4) adjustment of the outlier effect on the candidate exposures through multivariable MR. They test their framework on simulations and report four interesting examples of real data MR analyses where they report both well-known and promising novel causal exposure-outcome relationships using MR-TRYX. It a well-written and easy-to-follow manuscript.

Cho et al. tackle the "issue" of horizontal pleiotropy in an original and promising way. Exploiting horizontal pleiotropy to highlight new causal pathways is a very natural idea and next step in the MR field especially since they have the resources to do so through MR-Base. I still believe this is a very big challenge even if the authors proposal is very interesting and have a few questions and comments about the framework and its validation.

Major comments

METHOD

- To determine the pool of candidate exposures, why use IVW (and excluding the outlier SNP) and

not directly multivariable MR with both the hypothesized exposure and the candidate exposure to select candidate exposures "causal" to the outcome conditional on the hypothesized exposure?

- LASSO provides the best estimator in terms of MSE, therefore in terms of prediction, thus it introduced a bias to decrease the variance. However, the aim in MR is to have the best estimator in terms of unbiased causal estimate, not in terms of prediction. How does the LASSO selection step influence the bias in the causal estimate?
- I believe LASSO is very sensitive and the selection procedure is not very stable. For instance, I wouldn't be surprised to see the set of selected candidate exposures be affected by the clumping procedure. How stable is the LASSO selection procedure in MR-TRYX? Does the LASSO procedure favor some candidate exposures, such as exposures with many instruments? For instance, in example 1 of the real data, Height has been selected as a candidate exposure for CHD, wouldn't body mass index/weight/hip circumference be more relevant? Was BMI part of the original pool of candidate traits? If so, was Height selected by LASSO because Height has more instruments and hence provides a better predicted outcome (from the MSE standpoint) even if it is not the "causal" exposure?
- On a related note, would there be any benefit in having a step selection procedure where the instruments associated with unselected exposures at step i are removed at step $i+1$.
- To free MR-TRYX of the disadvantage of the outlier detection step, could the whole MR-TRYX framework be applied to all the instruments? Would the results differ much? My intuition would be that the results would be very similar in case the selection of candidate exposure is done using a conditional model on the hypothesized exposure (multivariable MR instead of IVW), is that a reasonable assumption?
- Since MR-TRYX is fully dependent on MR-Base, how does MR-Base deal with ancestry? with sample overlap?
- Instead of manually removing redundant traits, especially traits similar to the outcome, would it be a good idea to use genetic correlations using LD score?

SIMULATIONS

- I feel like the simulation design could be extended to span a larger range of scenarios.
- Most of the simulation scenarios are at a disadvantage for outlier-based method since the proportion of SNPs with pleiotropic effects is mostly high (6/8 scenarios with at least 30% of pleiotropic SNPs). In addition, the total number of SNPs, 30, is quite low so any method removing outliers automatically suffers from a lack of power. Therefore, I feel like it is hard to give it a fair comparison.
- I didn't find any mention of the direction of the pleiotropic effects which is of importance for outlier-based methods.
- It is not clear how big the pool of potential candidate exposures is and if which part of the MR-TRYX framework is run in the simulations. Is the LASSO step tested? Are the candidate exposures considered as known? If so it favors MR-TRYX.
- It would be interesting to see how the multivariable MR with the hypothesized exposure + the candidate exposure(s) (meaning without adjusting for outliers) compares to the tested methods. Is it perfectly equivalent to adjusting for outliers in MR-TRYX?
- In the real data, the authors use the reduction in Q statistic to assess how successful the MR-TRYX outlier adjustment is, but I have the intuition that any adjustment should reduce the Q statistic (by including an additional variable in a linear model, the residual variance decreases). How does the Q statistic compare between removing the outliers and adjusting for outliers? So, it would be nice to have such quantitative results in the simulations.

RESULTS

- The 4 examples highlighted by the authors are relevant and of interest however, we are lacking more global results insofar as possible. For instance, among a certain set of exposure-outcome traits, how often are outliers detected / are candidate exposures detected? What is the average number of candidate exposures per outlier SNP / per hypothesized exposure-outcome pair? How many new exposures can be discovered using MR-TRYX?
- Reduction in the Q statistic is a way for the authors to assess the success of their adjustment,

however, I believe that when adjusting for an additional variable in a linear model setting, the Q statistic automatically decreases. How "significant" are these observed reductions in the Q statistic value? Is heterogeneity still found after MR-TRYX adjustment, is Q still significant?

Minor comments

- I149: did the authors mean figure 1b?
- typo? I222: "To perform multivariable MR, more SNPs were introduced into the analysis that instrument the candidate traits.", pardon my misunderstanding of the sentence, do the authors mean "more SNPs, that are instruments for the candidate traits, were introduced into the analysis"?
- Figure 1: Number SNPs represented in b) and c) is different
- Figure 2: it would be nice to specify the total number of SNPs (30) somewhere in the x-axis label or having proportions.

Title: MR-TRYX: Exploiting horizontal pleiotropy to infer novel causal pathways

We appreciate the reviewers' critical comments and constructive suggestions. We have carefully addressed the comments and changed the manuscript accordingly. We hope that the revised manuscript has adequately addressed the reviewers' comments.

Reviewer #1

Major comments

1. My main comment is on how reliable the last part of the MR-TRYX approach – adjusting the MR estimates – is. In particular, it relies on the strong assumption that you can obtain an unbiased estimate of all of the relevant candidate traits P on the outcome Y. To their credit, on page 12 and in the discussion section the authors discuss many caveats and are upfront about the potential limitations. However, I feel the authors could be even more prudent and present the last part of the analysis as a way to tentatively gauge the direction of bias in the original MR estimate rather than presenting it as a bias-corrected MR estimate. MR-base is an extremely rich resource, and is expanding, but the MR-TRYX approach relies heavily on candidate traits being measurable and for which GWAS results exist. Therefore I feel a bit like you're back in the world of observational studies with unmeasured confounding: you're never sure whether you've adjusted for all the relevant candidate traits that bias the effect of X on Y.

Answer) As the reviewer kindly noted, we are very open about the caveats of the method. Every bias-corrected method for MR fails if its assumption of either knowledge of genetic architecture of the instruments (e.g. Egger, median, mode, outlier removal etc), or knowledge of the pleiotropic pathways (multivariable MR) is wrong. The bias-adjustment aspect of MR-TRYX takes elements from the former to make improvements on the latter, and we believe will be of value to researchers to be used alongside existing tools. We agree that it is difficult to conclude that we've adjusted for all pleiotropic traits, and the analogy you make to the problem of unmeasured confounding in observational associations is important. We have added an extra section in the discussion that addresses this point in detail (Page 23, line 508-519), and we have also added the following text at the first mention of the outlier-adjustment method in the 'Overview of MR-TRYX' section (Page 3, line 60-63):

“This outlier-adjustment method should be treated as a new approach to be used in conjunction with other methods that already exist in the MR sensitivity analysis toolkit. We provide extensive discussion on the context, advantages and potential pitfalls that come with trying to use a data-driven approach to adjust for horizontal pleiotropy at the end of the paper.”

We have now also run substantially more simulation scenarios and included multivariable MR as one of the method comparisons to more properly evaluate the performance of outlier

removal and outlier adjustment. As one might expect, there is no method that performs well in all scenarios, and we make it very clear that such is the case in the results section describing the simulations. But we do feel that the outlier adjustment strategy that we put forward here is a potentially useful contribution.

The reviewer also rightly points out that adjustment depends on the availability of data on the pleiotropic pathway. We have re-emphasised this, and we have also shown that if such data is not available, the method performs identically to random effects IVW, which itself performs relatively well across the range of scenarios (in fact, better than outlier removal).

One idea that came to mind is whether you could use negative controls or MRGxE to have some sense whether the adjustment through observable candidate traits is sufficient to eliminate bias. I mean, one way to purge the effect of X on Y from possible pleiotropic effects is to measure the relevant pleiotropic effects through observed candidate traits as the authors suggest.

An alternative approach is to purge the effect of X on Y from pleiotropic effects using no-relevance groups for whom $X=0$ by definition as in Chen et al. (2008), Van Kippersluis & Rietveld (2018), and Spiller et al. (2019). If both methods suggest a similar correction of the original IVW estimate, this raises confidence in the correction part of MR-TRYX.

Answer) Thank you for these suggestions. Triangulating amongst different methods is certainly an important strategy, and we agree that using GxE interactions for negative controls is a promising alternative approach. We just need to point out that there are three reasons that this is somewhat difficult in the context here: 1) Identifying robust GxE interactions for the instruments is very difficult; 2) Doing this in a 2-sample framework which is how the empirical analyses were performed with MR-TRYX is currently not possible with available data; 3) We believe that the empirical analyses are not actually in need of support from such an alternative method – there are no drastic changes to the point estimates from other methods the main point that we make is that we explain a large proportion of the heterogeneity. An important high-level empirical result is that most pleiotropy that we identified in the empirical analyses that we performed appeared to be balanced which leads to relatively low bias [1, 2].

References

[1] Sanderson E, Macdonald-Wallis C, Davey Smith G. Negative control exposure studies in the presence of measurement error: implications for attempted effect estimate calibration. *Int J Epidemiol.* 2018;47(2):587-596.

[2] Taylor AE, Davey Smith G, Munafò MR. Re: "Exposure to maternal smoking during pregnancy as a risk factor for tobacco use in adult offspring". *Am J Epidemiol.*

2. My second main comment is that I believe the authors should talk more about the hypothesized data generating process under which their approach works. To me it seems that the approach works best when one has an additive model in which X and all the candidate traits P have additive effects, and where none of the candidate traits P is a mediator or collider in the relationship between X and Y. But what if P is a mediator of the relationship between X and Y? What if P is a collider? What if there are interaction effects between X and P?

Answer) Thank you for these suggestions, they were very helpful. We have now substantially expanded the section describing the data generating process used in the simulations (Page 7, line 156-257), and this includes instances of P being a confounder of X and Y, P being a mediator between X and Y, and also just a straightforward mediator of horizontal pleiotropy. If P were a collider it wouldn't show a causal effect on Y, and hence would not be used for estimating the adjusted effect [1]. At this point the simulations are very extensive, more so than is often seen in methodology papers. Exploring interaction effects will quite substantially increase the parameter space of possible simulations and to do it justice we believe that we should treat this separately – not specifically related to MR-TRYX but related to all pleiotropy robust methods, as it is not something that has been adequately addressed in the literature so far.

References:

[1] Sanderson E, Davey Smith G, Windmeijer F, Bowden J. An examination of multivariable Mendelian randomization in the single-sample and two-sample summary data settings. *Int J Epidemiol.* 2018. [Epub ahead of print]

Related, redundancy of candidate traits is currently determined by a statistical approach (LASSO) but shouldn't redundancy not also be based on some theory or at least some idea of how the candidate trait relates to X and Y in e.g., a Directed Acyclical Graph (DAG)?

Answer) Thanks for the suggestion, this is actually an aspect of data management with which we continue to try to improve in the MR-Base framework. In data not shown, we actually tried many different approaches to overcome the problem of redundancy of data including PCA, clustering techniques, annotating to ontologies and pruning via the ontological trees, and data driven approaches. We remain unable to find a suitable automated method for avoiding the problem of erroneously introducing a 'candidate trait' which is a surrogate for Y in the adjustment analysis. As a consequence, we have alerted the reader that at this stage it is very important that they manually inspect the adjustment traits and to use human knowledge to avoid surrogates for Y being used to adjust the effect, as this will just nullify the entire model.

On the other hand, the LASSO approach appears to be very effective at removing surrogates

for X, and redundancy amongst the candidate traits. Incidentally, it appears to be almost identical in performance (though much faster) to another recently developed method [1].

We have performed additional simulations to evaluate the efficacy of the lasso approach in removing redundancy and its performance appears to be quite positive. There may be improvements on this element of the method in the future, but at this point we cannot find a more suitable way to avoid redundancy. We have provided different options in the software on how to deal with redundancy and have now written extensively about this in Supplementary note 1.

References

[1] Verena Zuber et al. Selecting causal risk factors from high-throughput experiments using multivariable Mendelian randomization. bioRxiv. 2019 [https://www.biorxiv.org/content/10.1101/396333v2]

3. I think the description of multivariable MR is a bit confusing. In the introduction it is mentioned that the causal influence of the candidate trait on the outcome is obtained using MR excluding the original outlier; then on page 7 the authors talk about a set of T clumped instruments for both X and P; and then finally on page 10 it is mentioned that additional SNPs are introduced to instrument the candidate traits. Please fill me in on how exactly the multivariable MR is performed.

Answer) Thank you for alerting us to the lack of clarity here. We have revised and largely rewritten the sections that you mentioned, including the legend of figure 1 and the methods section describing the candidate trait adjustment process. We have entirely re-written the section relating to redundant traits (Page 5-6) and moved it to supplementary note 1 in order to reduce confusion between that practical aspect of applied analysis and the core idea of the method.

4. Two comments on the selection of candidate traits and the differential use of false discovery rates:

- 1) Candidate traits are defined as those having an association with the outlier SNP at genome-wide significance level. However, since the pvalue in the GWAS is a function of both the effect size but also the sample size, this way of selecting traits is partly on basis of the accidental sample size of the discovery GWAS and not on basis of the potential influence on the MR estimate under study. Would it not be better to select traits on basis of the effect size (and hence the potential influence on the MR estimate)?**

Answer) Thanks for your suggestion, which we do think is a good point. However, there are a few reasons that we opted against using effect size as a threshold. First, the traits in MR-Base are not in standardised units because often the meta data relating to the original standard deviations of the analysed traits are not available, so currently it is not technically possible to treat all traits equally on the basis of effect size (just as we can't treat all traits equally on the basis of p-value as you have pointed out). Second, the relevance of a candidate trait in being on the outlier pathway is probably more strongly related to variance explained than just effect size, which includes the need for allele frequencies in addition to units and is hard to estimate for binary traits. We reasoned that if the candidate trait is relevant for the outlier then it seems implausible that the outlier would not reach GWAS significance for that trait, and therefore set that as the default. However, users can change this threshold. As the meta-data in the MR-Base database becomes more complete and we can harmonise all datasets to have uniform units across studies we will introduce the option for users to set thresholds based on R^2 or effect size thresholds rather than p-values.

- 2) Then candidate traits are further reduced by looking at the effect of these traits on the outcome. Here a p-value of 0.05 is used. In the outlier detection part a p-value of 0.05 divided by the number of SNPs is used as a correction for multiple testing. For consistency, wouldn't it be better to apply a similar correction to the 0.05 FDR here too?**

Answer) At the point of making adjustments, we feel that thresholds don't particularly matter, because if the effect estimate is very small or imprecise then the adjustment will be negligible. The arbitrary threshold of 0.05 is there because with a view to impose some nominal threshold for viewing potential new risk factors for the outcome, but again we must emphasise that this is a datamining exercise which necessitates p-value thresholds for convenience, and we don't necessarily advocate that they should not be overinterpreted.

The default outlier detection threshold is Bonferroni corrected because if there are a large number of instruments then we always expect some SNPs to be nominally significant as outliers, which is slightly in opposition to the notion that outliers are being driven by horizontal pleiotropy. i.e. they would not really be outliers. It also is computationally advantageous to set the default at this threshold to avoid server issues, though again, users are fully able to adjust these settings.

5. I don't find Figure 2 very illuminating. Could you also present the actual estimates in a Table such that readers can judge the bias? How is "substantially different" on line 665 defined?

Answer) Figure 2 is now completely changed having overhauled the simulations, and the actual estimates are now in supplementary table S5. We would prefer to keep Figure 2 as the

main presentation as the bias values themselves are not meaningful – they are based on chosen parameters that may or may not represent real situations (as we cannot easily know the true parameters in any particular analysis). Rather, the importance is to be able to see how within each scenario the different methods compare, and between scenarios the different methods perform inconsistently. We have emphasised this in the results section on the simulations now (Page 7-12).

We wrestled with how best to illustrate the bias in these simulations. Normally one would just present the average bias across a number of repeated simulations, but the problem with this is that we are not simulating directional pleiotropy, we are allowing the outliers to go in any direction, so across many simulations the average bias will tend towards 0 even if all individual simulations are themselves biased in either direction. In Figure 2 we are instead presenting the proportion of the simulations that are ‘significantly’ biased, i.e. has an effect estimate with CI not overlapping the true simulated effect. We believe this is the most appropriate way to depict these scenarios. Other ways are to show e.g. the mean squared error or the average absolute bias, but these are not easily interpretable, so we opted against those options.

Minor comments

1. **On page 1, line 56, you talk about ‘outliers’, but at this stage in the introduction it is not clearly defined what you mean by outliers. Perhaps you can help the reader by defining outliers as ‘suspicious SNPs’ or ‘SNPs exhibiting possible pleiotropic effects’?**
Answer) Revised.

2. **Page 4, lines 111-113, this part is very hard to understand for non-experts. Could you provide a bit more intuition or at least a good reference as you did before on line 102.**
Answer) We realise that these lines were needless repetitions of a previous sections (although written less clearly) and have now been removed. We added a little extra explanation in the first mention (Page 3, Line 66-80).

3. **In empirical example 1, isn’t ibuprofen use a potential collider?**

Answer) It could be a potential collider as people who are prescribed aspirin, which is used to treat myocardial infarction, might be advised to avoid chronic ibuprofen use. However, it has been reported that ibuprofen may affect cardiovascular risk [1]. Therefore, we assumed that ibuprofen can be a putative risk factor for cardiovascular disease rather than a collider. Also, the estimate was obtained from MVMR, which is not subject to collider bias (see the answer to #Reviewer 1. Q-2).

References

[1] Cross PL. Effect of ibuprofen on cardioprotective effect of aspirin. Lancet. 2003 May 3;361(9368):1560-1.

- 4. To fairly compare the confidence intervals across methods, and given that 1000 bootstrap replications is fair but not very large, would it be useful to also bootstrap the standard errors of the IVW approach under the same number of replications, to make sure that the differences in confidence intervals are due to the method of estimation and not the method of computing standard errors?**

Answer) This is an interesting point, but we think it might be outside the scope of this paper. The method of determining SE of the IVW approach that we are comparing against is the standard one used in the field and we are interested in showing how that implementation compares against the one that we have developed here. One way in which we have improved on the problem that the reviewer has rightly pointed out is the following: Previously we were comparing false discovery rates in the null models and the power in the non-null models separately. So, if some method had a higher FDR it would potentially also have a higher power, which is not a meaningful result. Instead, we now evaluate the overall sensitivity and specificity of each method in discriminating between a null and a non-null model, by reporting the area under the receiver operator curve (Page 7-12; Figure 2).

- 5. I don't understand the sentence on line 434-435 on the shrinkage step.**

Answer) Although heterogeneity is bad when we test the validity of instrument, it is necessary when we assess the strength of instrument. Suppose that Q statistics for exposure X_1 and X_2 are Q_{X_1} and Q_{X_2} . Q statistics here indicates instrument relevance, which detects the level of heterogeneity to determine if a set of instruments can predict all exposures of interest. If both Q_{X_1} and Q_{X_2} are larger than the chosen critical value for X_2 distribution, detection of heterogeneity suggests that the instruments can predict variation in both exposures. As multivariable MR requires more than one trait, SNP-exposure effect heterogeneity is necessary to obtain valid estimates.

- 6. The legend and labels of the figures could be improved.**

Answer) Revised.

Reviewer #2

1. The authors selected a P-value threshold of 5×10^{-8} for selecting SNPs exhibiting horizontal pleiotropy in associations with secondary traits and outcomes, which may be too conservative especially given GWAS with smaller sample sizes.

Answer) Thanks for your comment. The use of conservative threshold helps to reduce the number of false-positive associations arising from the vast number of statistical tests performed. It also simplifies subsequent steps, because we need to select instruments for candidate traits, which are canonically chosen at GWAS significance level and ideally the outlier SNPs will have the same level of evidence. Nevertheless, the software allows the user to select their own thresholds.

2. It would be useful if the authors could add the number of genetic instruments in the Results for the four practical examples that they use.

Answer) We provided the number of genetic instruments used in the analysis in the Table 1 and 2.

Reviewer #3

Major comments

Thank you for your very helpful comments, we hope the paper has improved in light of them.

1. METHOD

1) To determine the pool of candidate exposures, why use IVW (and excluding the outlier SNP) and not directly multivariable MR with both the hypothesized exposure and the candidate exposure to select candidate exposures "causal" to the outcome conditional on the hypothesized exposure?

Answer) Multivariable MR estimates the direct effect of each exposure included in the model on the outcome. Therefore, if any of the effect on the exposure of interest is mediated by other exposures included this mediated effect will not appear as part of the direct effect. MR TRYX however is a univariable MR and estimates the total effect of the exposure of interest on the outcome, this will include any effect that is mediated through other potential exposures (including those that are affected by pleiotropic SNPs). MR TRYX removes the effect of pleiotropy in the conventional MR estimates whilst still estimating the total effect of the exposure on the outcome. We used IVW to estimate the total effect of the exposure of interest on the outcome. This is highlighted now in the simulations (Page 7-12) and Figure 2.

2) LASSO provides the best estimator in terms of MSE, therefore in terms of prediction, thus it introduced a bias to decrease the variance. However, the aim in MR is to have the best estimator in terms of unbiased causal estimate, not in terms of prediction. How does the LASSO selection step influence the bias in the causal estimate?

Answer) Thank you for this important point. We have substantially re-evaluated the LASSO step and agree that it is not necessarily the best estimated in terms of bias in all scenarios. The part of the method that uses LASSO is essentially there to deal with the practical problem of many redundant variables existing in the MR-Base database and we have hopefully made this much clearer (Page 5-7), as it has a large section dedicated to it in Supplementary note 1. Under the sufficient condition for identification that less than 50% of variables are redundant, the LASSO selection may select the valid traits. However, this step could influence the power of MVMR, which affects the validity of adjusted estimates. We described this potential limitation in the Discussion (Page 23, line 520-550) and Supplementary note 1. Additionally, we provided additional simulations (Page 7-12) to show the validity of the LASSO step (Please see #Reviewer 3; Q2-4), and additional methods in the software to account for the case when multiple candidate traits associate with an outlier.

3) I believe LASSO is very sensitive and the selection procedure is not very stable. For

instance, I wouldn't be surprised to see the set of selected candidate exposures be affected by the clumping procedure. How stable is the LASSO selection procedure in MR-TRYX? Does the LASSO procedure favor some candidate exposures, such as exposures with many instruments? For instance, in example 1 of the real data, Height has been selected as a candidate exposure for CHD, wouldn't body mass index/weight/hip circumference be more relevant? Was BMI part of the original pool of candidate traits? If so, was Height selected by LASSO because Height has more instruments and hence provides a better predicted outcome (from the MSE standpoint) even if it is not the "causal" exposure?

Answer) We are aware that there were issues with the original LARS implementation of LASSO but the stability of the glmnet implementation is good – in our simulations and empirical analyses we do not find instability issues.

We simulated whether the number of instruments for the candidate trait affect the ability of LASSO to appropriately select candidate traits.

[Additional simulation 1] In this additional simulation, we generated traits X1 and X2 which are instrumented by 100 SNPs and 20 SNPs, respectively. We set that 10 of them are pleiotropic and associated with both of X1 and X2. Four scenarios are considered:

- 1) X1 has an effect on Y ($\beta=0.3$), where X2 has no effect on Y ($\beta=0.0$)
- 2) X1 has no effect on Y ($\beta=0.0$), where X2 has an effect on Y ($\beta=0.3$)
- 3) Both of X1 and X2 have effects on Y ($\beta=0.3$)
- 4) Neither of X1 and X2 have effects on Y ($\beta=0.0$)

In each scenario, we performed the LASSO regression to see if the LASSO selects the traits in a manner that is determined by which trait is causal, rather than which trait has more instruments. The simulation was replicated 1000 times in each scenario. The results showed that the LASSO step worked well across the scenarios. The LASSO kept both X1 and X2 when both traits have effects on the outcome Y with a probability of 1.000 (1000 times per 1000 simulations), regardless of the number of instruments for the traits. Whilst, the LASSO removed both traits with a probability of 0.693 when both of X1 and X2 have no effects on Y. In this scenario, the trait X1 was removed with a probability of 0.775 where the trait X2 was removed with a probability of 0.783. In scenario 1, LASSO kept the trait X1 100% but failed to remove X2 sometimes, which is instrumented by a small number of variants and has no effect on Y, with a probability of 0.473. In scenario 2, X2 wasn't removed by the LASSO step but X1 was remained with a probability of 0.467. In conclusion, the results suggested that the LASSO regression is not preferentially selecting traits based on number of instruments, but the number of instruments may influence statistical power. We have added these simulations to the supplementary materials (Simulation 1 and 2 in Supplementary file).

4) On a related note, would there be any benefit in having a step selection procedure where the instruments associated with unselected exposures at step i are removed at

step i+1.

Answer) In terms of MVMR, MR-TRYX keeps only the instruments associated with the exposures (or candidate traits) that are involved in the analysis. Unselected candidate traits and related instruments are automatically excluded from the analysis.

5) To free MR-TRYX of the disadvantage of the outlier detection step, could the whole MR-TRYX framework be applied to all the instruments? Would the results differ much? My intuition would be that the results would be very similar in case the selection of candidate exposure is done using a conditional model on the hypothesized exposure (multivariable MR instead of IVW), is that a reasonable assumption?

Answer) It's an interesting suggestion, and we did originally mention it in the Discussion (Page 21 and 23). We have now added it as a scenario in the simulations, and we find that it doesn't perform as well overall as using outliers for adjustment. But it could certainly be useful for generating more hypotheses for potential influences on the outcome.

6) Since MR-TRYX is fully dependent on MR-Base, how does MR-Base deal with ancestry? with sample overlap?

Answer) As you pointed out, the association estimates can be biased when candidate traits and outcome studies are conducted in different populations. MR-Base provides meta-data on population characteristics [1], including ancestry, and geographic origin to guide the user in selecting the most appropriate design for their analysis. So far, the majority of studies are derived from European population, but we are expanding the scope of MR-Base by collecting summary statistics from various populations. Therefore, it is recommended to ensure that selected candidate traits and the outcome were obtained from the same population. We added this in the Discussion as follows (Page 24, Line 545-551):

“Fifth, since MR-TRYX uses the resource from MR-Base, it is recommended that the user acknowledge the limitation and restriction of MR-Base. For example, the population should be the same for the exposure (or the candidate traits) and the outcome traits to avoid mis-estimation of the magnitude of the association. The users should consider modifying their analyses when the limitation indicated above are avoidable.”

7) Instead of manually removed redundant traits, especially traits similar to the outcome, would it be a good idea to use genetic correlations using LD score?

Answer) Thanks for your suggestion. MR-TRYX was designed to create a user-friendly environment. User can use an automated approach for identifying candidate traits, but as part of sensitivity analysis, they can curate the pool of candidate traits manually according to their study hypothesis. Automatically detecting redundant traits is ongoing work, we attempted to

do this in many ways (data not shown) including using LD score regression, clustering by PCA, annotating to ontologies and then pruning ontological trees, and other approaches. None of them work sufficiently well to be used automatically, and so we think it's most prudent to do this manually at this stage.

2. SIMULATIONS

1) I feel like the simulation design could be extended to span a larger range of scenarios.

Answer) Thanks for the suggestion. We have now substantially expanded the simulation scenarios, including more effects, directional and balanced pleiotropy, full and partial mediating traits, and genetically influenced confounders.

2) Most of the simulation scenarios are at a disadvantage for outlier-based method since the proportion of SNPs with pleiotropic effects is mostly high (6/8 scenarios with at least 30% of pleiotropic SNPs).

Answer) There is growing evidence that horizontal pleiotropy is likely the predominant form of pleiotropy, but this remains an unsolved question. We cannot know what is realistic for the simulations, so the objective is to evaluate what scenarios would give rise to problems for different methods, not which methods would perform the best in the most likely scenarios. The results are now more balanced in their presentation, making what we feel is an important point that there is no method that performs well across all scenarios (MR-TRYX included), and that many methods that perform well in some scenarios perform very badly in others. The point is to urge caution against advocating the use of a single analytical method and to develop new methods that contribute to a growing toolkit of sensitivity analyses.

3) I didn't find any mention of the direction of the pleiotropic effects which is of importance for outlier-based methods.

Answer) We now substantially improved the methods section relating to the simulations (Page 7-11) and have also included both balanced and directional models of horizontal pleiotropy in the simulations.

4) It is not clear how big the pool of potential candidate exposures is and if which part of the MR-TRYX framework is run in the simulations. Is the LASSO step tested? Are the candidate exposures considered as known? If so it favors MR-TRYX.

Answer) Yes, great point. The simulations are already very extensive, and we will struggle to expand them to mimic the messiness of real data redundancy, while delivering an interpretable message. We have now re-worded much of the manuscript to make it clear that

MR-TRYX is a framework that brings together various techniques, and a novel technique that we introduce within that framework is adjusting for outliers. The purpose of the main simulation is now stated to evaluate the performance of outlier adjustment vs other methods. We have now also done additional separate simulations to evaluate the data redundancy aspect (Supplementary note 1).

The pool of potential candidate traits varies across the simulations as is now described in the methods. The number of candidate traits in the simulations is typically much smaller than what is available in MR-Base but selection is based on multiple testing thresholds related to how many traits are available and if candidate traits are not detected empirically then MR-TRYX just becomes IVW. One of the options for use of the software is to retain only a single candidate trait per SNP, which is in line with most of the simulation scenarios.

5) It would be interesting to see how the multivariable MR with the hypothesized exposure + the candidate exposure(s) (meaning without adjusting for outliers) compares to the tested methods. Is it perfectly equivalent to adjusting for outliers in MR-TRYX?

Answer) We have now included this method in the simulations. We describe how it is different to outlier adjustment – i.e. it adjusts for all instruments not just the outliers. As a consequence, there are scenarios in which it performs better and ones in which it performs much worse. For example, if the candidate trait partially or completely mediates the path from the exposure to the outcome then the effect of x is completely nullified, which is not a problem seen for MR-TRYX. Figure 2 now includes these results.

6) In the real data, the authors use the reduction in Q statistic to assess how successful the MR-TRYX outlier adjustment is, but I have the intuition that any adjustment should reduce the Q statistic (by including an additional variable in a linear model, the residual variance decreases). How does the Q statistic compare between removing the outliers and adjusting for outliers? So, it would be nice to have such quantitative results in the simulations.

Answer) Our adjustment model doesn't include an extra term in the revised X-Y association. Instead of adding variables, we correct the G-Y association by subtracting estimated pleiotropic effect obtained from LASSO MVMR from the total effect. The Q statistics derived from our method is not necessarily decreased as we could equally adjust in the correct or incorrect direction thereby increasing or decreasing heterogeneity. Indeed, when we adjust the pleiotropy in the wrong direction (Please see the example 4, where we acknowledge that in one empirical situation the adjustment is not working and goes in the wrong direction).

3. RESULTS

1) The 4 examples highlighted by the authors are relevant and of interest however, we

are lacking more global results insofar as possible. For instance, among a certain set of exposure-outcome traits, how often are outliers detected / are candidate exposures detected? What is the average number of candidate exposures per outlier SNP / per hypothesized exposure-outcome pair? How many new exposures can be discovered using MR-TRYX?

Answer) When multiple SNPs ($n > 1$) are used in MR analysis, it is possible that some variants are valid instruments, but others are not. The number of invalid instruments (so called as outliers) and candidate exposures can vary depending on the research hypothesis. The primary goal of this study is to suggest the idea to overcome the problem of pleiotropy in the current MR model and to open the possibility of a hypothesis free screen for potential exposures. In this paper, therefore, we didn't examine all possible exposure-outcome associations to let the users explore their research hypothesis where horizontal pleiotropy exists. It would also be very hard to do this systematically because there remains a manual step involved in the analysis. We believe that the 4 examples we provided are reasonably exemplars showing instances where potentially novel pathways are identified and some successes and failures of the new method (Page 19-20). Currently, 11 billion SNP-trait association from 1673 GWAS are available in MR-Base.

2) Reduction in the Q statistic is a way for the authors to assess the success of their adjustment, however, I believe that when adjusting for an additional variable in a linear model setting, the Q statistic automatically decreases. How "significant" are these observed reductions in the Q statistic value? Is heterogeneity still found after MR-TRYX adjustment, is Q still significant?

Answer) We resorted to evaluating success by the Q statistic because a) for the analyses that we selected we ultimately found that all methods resulted in quite similar effect estimates despite there being substantial heterogeneity, and b) the change in Q could quite easily go up or down if the method is not working or working - either we adjust the effects in the incorrect or correct direction. The new Q values are typically reduced but also typically remain significant, these results are outlined in Table 2.

Minor comments

1. 1149: did the authors mean figure 1b?

Answer) Sorry for the confusion. It has been revised.

2. typo? 1222: "To perform multivariable MR, more SNPs were introduced into the analysis that instrument the candidate traits.", pardon my misunderstanding of the

sentence, do the authors mean "more SNPs, that are instruments for the candidate traits, were introduced into the analysis"?

Answer) Sorry for the confusion. To convey clear message, this has been revised.

3. Figure 1: Number SNPs represented in b) and c) is different

Answer) Revised.

4. Figure 2: it would be nice to specify the total number of SNPs (30) somewhere in the x-axis label or having proportions.

Answer) Revised.

Reviewers' Comments:

Reviewer #1:

Remarks to the Author:

I think the authors did an excellent job in responding to my comments and in revising the text. I feel the new version of the manuscript is much improved in terms of clarity and scope, and recommend publication.

Reviewer #2:

Remarks to the Author:

The authors have adequately addressed my comments.

Reviewer #3:

Remarks to the Author:

I believe my concerns have now been adequately addressed. The paper is clearer and I appreciate the added results and comments from the authors which improve the paper in my opinion. I was particularly interested to read the additional results about the LASSO selection procedure.